# Differential privacy for eye tracking with temporal correlations

**Efe Bozkir**[ID][1☯]*, **Onur Günlü**[ID][2☯], **Wolfgang Fuhl**[1], **Rafael F. Schaefer**[ID][2], **Enkelejda Kasneci**[ID][1]

**1** Chair of Human-Computer Interaction, University of Tübingen, Tübingen, Germany, **2** Chair of Communications Engineering and Security, University of Siegen, Siegen, Germany

☯ These authors contributed equally to this work.
* efe.bozkir@uni-tuebingen.de

**Data Availability Statement:** Relevant data files are provided via following url: https://atreus.informatik.uni-tuebingen.de/bozkir/dp_eye_tracking.

## Abstract

New generation head-mounted displays, such as VR and AR glasses, are coming into the market with already integrated eye tracking and are expected to enable novel ways of human-computer interaction in numerous applications. However, since eye movement properties contain biometric information, privacy concerns have to be handled properly. Privacy-preservation techniques such as differential privacy mechanisms have recently been applied to eye movement data obtained from such displays. Standard differential privacy mechanisms; however, are vulnerable due to temporal correlations between the eye movement observations. In this work, we propose a novel transform-coding based differential privacy mechanism to further adapt it to the statistics of eye movement feature data and compare various low-complexity methods. We extend the Fourier perturbation algorithm, which is a differential privacy mechanism, and correct a scaling mistake in its proof. Furthermore, we illustrate significant reductions in sample correlations in addition to query sensitivities, which provide the best utility-privacy trade-off in the eye tracking literature. Our results provide significantly high privacy without any essential loss in classification accuracies while hiding personal identifiers.

## Introduction

Recent advances in the field of head-mounted displays (HMDs), computer graphics, and eye tracking enable easy access to pervasive eye trackers along with modern HMDs. Soon, the usage of such devices might result in a significant increase in the amount of eye movement data collected from users across different application domains such as gaming, entertainment, or education. A large part of this data is indeed useful for personalized experience and user-adaptive interaction. Especially in virtual and augmented reality (VR/AR), it is possible to derive plenty of sensitive information about users from the eye movement data. In general, it has been shown that eye tracking signals can be employed for activity recognition even in challenging everyday tasks [1–3], to detect cognitive load [4, 5], mental fatigue [6], and many other user states. Similarly, assessment of situational attention [7], expert-novice analysis in areas

**Funding:** O. Günlü and R. F. Schaefer are supported by the German Federal Ministry of Education and Research (BMBF) within the national initiative for "Post Shannon Communication (NewCom)" under the Grant 16KIS1004. We acknowledge support by Open Access Publishing Fund of University of Tübingen. The funders had no role in study design, data collection and analysis, decision to publish, or preparation of the manuscript.

**Competing interests:** The authors have declared that no competing interests exist.

such as medicine [8] and sports [9], detection of personality traits [10], and prediction of human intent during robotic hand-eye coordination [11] can also be carried out based on eye movement features. Additionally, eye movements are useful for early detection of anomias [12] and diseases [13]. More importantly, eye movement data allow biometric authentication, which is considered to be a highly sensitive task [14]. A task-independent authentication using eye movement features and Gaussian mixtures is, for example, discussed by Kinnunen et al. [15]. Additionally, biometric identification based on an eye movements and oculomotor plant model are introduced by Komogortsev and Holland [16] and by Komogortsev et al. [17]. Eberz et al. [18] discuss that eye movement features can be used reliably also for authentication both in consumer level devices and various real world tasks, whereas Zhang et al. [19] show that continuous authentication using eye movements is possible in VR headsets. While authentication via eye movements could be useful in biometric applications, the applications that do not require any authentication step possess privacy risks for the individuals if such information is not hidden in the data. In addition, if such information is linked to personal identifiers, the risk might be even higher.

As biometric content can be retrieved from eye movements, it is important to protect them against adversarial behaviors such as membership inference. According to Steil et al. [20, p. 3], people agree to share their eye tracking data if a governmental health agency is involved in owning data or if the purpose is research. Therefore, privacy-preserving techniques are needed especially on the data sharing side of eye tracking considering that the usage of VR/AR devices with integrated eye trackers increases. As removing only the personal identifiers from data is not enough for anonymization due to linkage attacks [21], more sophisticated techniques for achieving user level privacy are necessary. Differential privacy [22, 23] is one effective solution, especially in the area of database applications. It protects user privacy by adding randomly generated noise for a given sensitivity and desired privacy parameter, $\epsilon$. The differentially private mechanisms provide aggregate statistics or query answers while protecting the information of whether an individual's data was contained in a dataset. However, high dimensionality of the data and temporal correlations can reduce utility and privacy, respectively. Since eye movement features are high dimensional, temporally correlated, and usually contain recordings with long durations, it is important to tackle utility and privacy problems simultaneously. For eye movement data collected from HMDs or smart glasses, both local and global differential privacy can be applied. Applying differential privacy mechanisms to eye movement data would optimally anonymize the query outcomes that are carried out on such data while keeping data utility and usability high enough. As opposed to global differential privacy, local differential privacy adds user level noise to the data but assumes that the user sends data to a central data collector after adding local noise [24, 25]. While both could be useful depending on the application use-case, for this work, we focus on global differential privacy, considering that in many VR/AR applications which collect eye movement data, there is a central trusted user-level data collector and publisher.

To apply differential privacy to the eye movement data, we evaluate the standard Laplace Perturbation Algorithm (LPA) [22] of differential privacy and Fourier Perturbation Algorithm (FPA) [26]. The latter is suitable for time series data such as the eye movement feature signals. We propose two different methods that apply the FPA to chunks of data using original eye movement feature signals or consecutive difference signals. While preserving differential privacy using parallel compositions, chunk-based methods decrease query sensitivity and computational complexity. The difference-based method significantly decreases the temporal correlations between the eye movement features in addition to the decorrelation provided by the FPA that uses the discrete Fourier transform (DFT) as, e.g., in the works of Günlü and İşcan [27] and Günlü et al. [28]. The difference-based method provides a higher level of

privacy since consecutive sample differences are observed to be less correlated than original consecutive data. Furthermore, we evaluate our methods using differentially private eye movement features in document type, gender, scene privacy sensitivity classification, and person identification tasks on publicly available eye movement datasets by using similar configurations to previous works by Steil et al. [20, 29]. To generate differentially private eye movement data, we use the complete data instead of applying a subsampling step, used by Steil et al. [20] to reduce the sensitivity and to improve the classification accuracies for document type and privacy sensitivity. In addition, the previous work [20] applies the exponential mechanism for differential privacy on the eye movement feature data. The exponential mechanism is useful for situations where the best enumerated response needs to be chosen [30]. In eye movements, we are not interested in the "best" response but in the feature vector. Therefore, we apply the Laplace mechanism. In summary, we are the first to propose differential privacy solutions for aggregated eye movement feature signals by taking the temporal correlations into account, which can help provide user privacy especially for HMD or smart glass usage in VR/AR setups.

Our main contributions are as follows. (1) We propose chunk-based and difference-based differential privacy methods for eye movement feature signals to reduce query sensitivities, computational complexity, and temporal correlations. Furthermore, (2) we evaluate our methods on two publicly available eye movement datasets, i.e., MPIIDPEye [20] and MPIIPrivacEye [29], by comparing them with standard techniques such as LPA and FPA using the multiplicative inverse of the absolute normalized mean square error (NMSE) as the utility metric. In addition, we evaluate document type and gender classification, and privacy sensitivity classification accuracies as classification metrics using differentially private eye movements in the MPIIDPEye and MPIIPrivacEye datasets, respectively. Classification accuracy is used in the literature as a practical utility metric that shows how useful the data and proposed methods are. Our utility metric also provides insights into the divergence trend of differentially private outcomes and is analytically trackable unlike the classification accuracy. For both datasets, we also evaluate person identification task using differentially private data. Our results show significantly better performance as compared to previous works while handling correlated data and decreasing query sensitivities by dividing the data into smaller chunks. In addition, our methods hide personal identifiers significantly better than existing methods.

## Previous research

There are few works that focus on privacy-preserving eye tracking. Liebling and Preibusch [31] provide motivation as to why privacy considerations are needed for eye tracking data by focusing on gaze and pupillometry. Practical solutions are; therefore, introduced to protect user identity and sensitive stimuli based on a degraded iris authentication through optical defocus [32] and an automated disabling mechanism for the eye tracker's ego perspective camera with the help of a mechanical shutter depending on the detection of privacy sensitive content [29]. Furthermore, a function-specific privacy model for privacy-preserving gaze estimation task and privacy-preserving eye videos by replacing the iris textures are proposed by Bozkir and Ünal et al. [33] and by Chaudhary and Pelz [34], respectively. In addition, solutions for privacy-preserving eye tracking data streaming [35] and real-time privacy control for eye tracking systems using area-of-interests [36] are also introduced in the literature. These works lack studying effects of temporal correlations.

For the user identity protection on aggregated eye movement features, works that focus on differential privacy are more relevant for us. Recently, standard differential privacy mechanisms are applied to heatmaps [37] and eye movement data that are obtained from a VR setup

[20]. These works do not address the effects of temporal correlations in eye movements over time in the privacy context. In the privacy literature, there are privacy frameworks such as the Pufferfish [38] or the Olympus [39] for correlated and sensor data, respectively. These works, however, have different assumptions. For instance, the Pufferfish requires a domain expert to specify potential secrets and discriminative pairs, and Olympus models privacy and utility requirements as adversarial networks. As our focus is to protect user identity in the eye movements, we opt for differential privacy by discussing the effects of temporal correlations in eye movements over time and propose methods to reduce them. It has been shown that standard differential privacy mechanisms are vulnerable to temporal correlations as such mechanisms assume that data at different time points are independent from each other or adversaries lack the information about temporal correlations, leading an increased privacy loss of a differential privacy mechanism over time due to the temporal correlations [40, 41]. The aggregated eye movement features over time might end up in an extreme case of such correlations due to various user behaviors. Therefore, in addition to discussing the effects of such correlations on differential privacy over time, we propose methods to reduce the correlations so that the privacy leakage due to the temporal correlations are minimal.

## Materials and methods

In this section, the theoretical background of differential privacy mechanisms, proposed methods, and evaluated datasets are discussed.

### Theoretical background

Differential privacy uses a metric to measure the privacy risk for an individual participating in a database. Considering a dataset with weights of $N$ people and a mean function, when an adversary queries the mean function for $N$ people, the average weight over $N$ people is obtained. After the first query, an additional query for $N - 1$ people automatically leaks the weight of the remaining person. Using differential privacy, noise is added to the outcome of a function so that the outcome does not significantly change based on whether a randomly chosen individual participated in the dataset. The amount of noise added should be calibrated carefully since a high amount of noise might decrease the utility. We next define differential privacy.

**Definition 1**. $\epsilon$-Differential Privacy ($\epsilon$-DP) [22, 23]. *A randomized mechanism M is $\epsilon$-differentially private if for all databases D and D′ that differ at most in one element for all S ⊆ Range(M) with*

$$\Pr\left[M(D) \in S\right] \leq e^{\epsilon} \Pr\left[M(D') \in S\right]. \tag{1}$$

The variance of the added noise depends on the query sensitivity, which is defined as follows.

**Definition 2**. Query sensitivity [22]. *For a random query $X^n$ and $w \in \{1, 2\}$, the query sensitivity $\Delta_w$ of $X^n$ is the smallest number for all databases D and D′ that differ at most in one element such that*

$$|| X^n(D) - X^n(D') ||_w \leq \Delta_w(X^n) \tag{2}$$

*where the $L_w$-distance is defined as*

$$|| X^n ||_w = \sqrt[w]{\sum_{i=1}^{n} \left(|X_i|\right)^w}. \tag{3}$$

We list theorems that are used in the proposed methods.

**Theorem 1**. *Sequential composition theorem* [42]. *Consider n mechanisms $M_i$ that randomization of each query is independent for $i = 1, 2, \ldots, n$. If $M_1, M_2, \ldots, M_n$ are $\epsilon_1, \epsilon_2, \ldots, \epsilon_n$-differentially private, respectively, then their joint mechanism is $\left(\sum_{i=1}^{n} \epsilon_i\right)$-differentially private.*

**Theorem 2**. *Parallel composition theorem* [42]. *Consider n mechanisms as $M_i$ for $i = 1, 2, \ldots, n$ that are applied to disjoint subsets of an input domain. If $M_1, M_2, \ldots, M_n$ are $\epsilon_1, \epsilon_2, \ldots, \epsilon_n$-differentially private, respectively, then their joint mechanism is $\left(\max_{i \in [1,n]} \epsilon_i\right)$-differentially private.*

We define the Laplace Perturbation Algorithm (LPA) [22]. To guarantee differential privacy, the LPA generates the noise according to a Laplace distribution. $Lap(\lambda)$ denotes a random variable drawn from a Laplace distribution with a probability density function (PDF): $\Pr[Lap(\lambda) = h] = \frac{1}{2\lambda}e^{-|h|/\lambda}$, where $Lap(\lambda)$ has zero mean and variance $2\lambda^2$. We denote the noisy and differentially private values as $\tilde{X}_i = X_i(D) + Lap(\lambda)$ for $i = 1, 2, \ldots, n$. Since we have a series of eye movement observations, the final noisy eye movement observations are generated as $\tilde{X}^n = X^n(D) + Lap^n(\lambda)$, where $Lap^n(\lambda)$ is a vector of $n$ independent $Lap(\lambda)$ random variables and $X^n(D)$ is the eye movement observations without noise. The LPA is $\epsilon$-differentially private for $\lambda = \Delta_1(X^n)/\epsilon$ [22].

We define the error function that we use to measure the differences between original $X^n$ and noisy $\tilde{X}^n$ observations. For this purpose, we use the metric normalized mean square error (NMSE) defined as

$$\text{NMSE} = \frac{1}{n}\sum_{i=1}^{n}\frac{(X_i - \tilde{X}_i)^2}{\overline{X}\overline{\tilde{X}}} \qquad (4)$$

where

$$\overline{X} = \frac{1}{n}\sum_{i=1}^{n}X_i, \qquad \overline{\tilde{X}} = \frac{1}{n}\sum_{i=1}^{n}\tilde{X}_i. \qquad (5)$$

We define the utility metric as

$$\text{Utility} = \frac{1}{|\text{NMSE}|}. \qquad (6)$$

As differential privacy is achieved by adding random noise to the data, there is a utility-privacy trade-off. Too much noise leads to high privacy; however, it might also result in poor analyses on the further tasks on eye movements. Therefore, it is important to find a good trade-off.

## Methods

Standard differential privacy mechanisms are vulnerable to temporal correlations, since the independent noise realizations that are added to temporally correlated data could be useful for adversaries. However, decorrelating the data without the domain knowledge before adding the noise might remove important eye movement patterns and provide poor results in analyses. Many eye movement features are extracted by using time windows, as in previous work [20, 29], which makes the features highly correlated. Another challenge is that the duration of eye tracking recordings could change depending on the personal behaviors, skills, or personalities of the users. The longer duration causes an increased query sensitivity, which means that higher amounts of noise should be added to achieve differential privacy. In addition, when

correlations between different data points exist, $\epsilon'$ is defined as the actual privacy metric instead of $\epsilon$ [43] that is obtained considering the fact that correlations can be used by an attacker to obtain more information about the differentially private data by filtering. In this work, we discuss and propose generic low-complexity methods to keep $\epsilon'$ small for eye movement feature signals. To deal with correlated eye movement feature signals, we propose three different methods: FPA, chunk-based FPA (CFPA) for original feature signals, and chunk-based FPA for difference based sequences (DCFPA). The sensitivity of each eye movement feature signal is calculated by using the $L_w$-distance such that

$$
\begin{aligned}
\Delta_w^f(X^n) &= \max_{p,\,q} \|X^{n,(p,f)} - X^{n,(q,f)}\|_w \\
&= \max_{p,\,q} \sqrt[w]{\sum_{t=1}^{n} (|X_t^{(p,f)} - X_t^{(q,f)}|)^w}
\end{aligned}
\tag{7}
$$

where $X^{n,\,(p,f)}$ and $X^{n,\,(q,f)}$ denote observation vectors for a feature $f$ from two participants $p$ and $q$, $n$ denotes the maximum length of the observation vectors, and $w \in \{1, 2\}$.

**Fourier Perturbation Algorithm (FPA).** In the FPA [26], the signal is represented with a small number of transform coefficients such that the query sensitivity of the representative signal decreases. A smaller query sensitivity decreases the noise power required to make the noisy signal differentially private. In the FPA, the signal is transformed into the frequency domain by applying Discrete Fourier Transform (DFT), which is commonly applied as a non-unitary transform. The frequency domain representation of a signal consists of less correlated transform coefficients as compared to the time domain signal due to the high decorrelation efficiency of the DFT. Therefore, the correlation between the eye movement feature signals is reduced by applying the DFT. After the DFT, the noise sampled from the LPA is added to the first $k$ elements of $DFT(X^n)$ that correspond to $k$ lowest frequency components, denoted as $F^k = DFT^k(X^n)$. Once the noise is added, the remaining part (of size $n - k$) of the noisy signal $\tilde{F}^k$ is zero padded and denoted as $PAD^n(\tilde{F}^k)$. Lastly, using the Inverse DFT (IDFT), the padded signal is transformed back into the time domain. We can show that $\epsilon$-differential privacy is satisfied by the FPA for $\lambda = \frac{\sqrt{n}\sqrt{k}\Delta_2(X^n)}{\epsilon}$ unlike the value claimed in previous work [26], as observed independently by Kellaris and Papadopoulos [44]. The procedure is summarized in Fig 1, and the proof is provided below. Since not all coefficients are used, in addition to the perturbation error caused by the added noise, a reconstruction error caused by the lossy compression is introduced. It is important to determine the number of used coefficients $k$ to minimize the total error. We discuss how we choose $k$ values for FPA-based methods below.

**Proof of FPA being differentially private.** We next prove that the FPA is $\epsilon$ differentially private for $\lambda = (\sqrt{n}\sqrt{k}\Delta_2(X^n))/\epsilon$. First, we prove the inequalities $(a)$ and $(b)$ in the following.

$$
\Delta_1(\hat{F}^n) \overset{(a)}{\leq} \sqrt{k} \cdot \Delta_2(\hat{F}^n) \overset{(b)}{\leq} \sqrt{n} \cdot \sqrt{k} \cdot \Delta_2(X^n)
\tag{8}
$$

where $\hat{F}^n(I) = [\hat{F}^k(I), 0, 0, \ldots, 0]$ such that $n - k$ zeros are padded. Consider $(8)(a)$, which

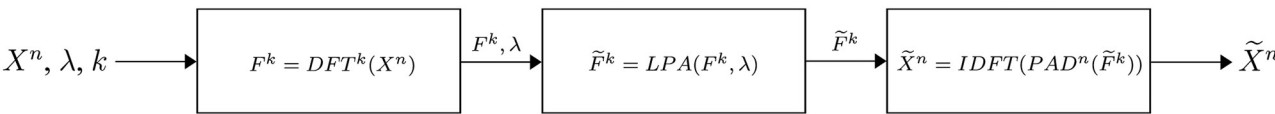

**Fig 1. Flow of the Fourier Perturbation Algorithm (FPA).**

follows since we have

$$
\begin{aligned}
\Delta_1(\hat{F}^n) &= \max_{I, I'} \| \hat{F}^n(I) - \hat{F}^n(I') \|_1 = \max_{I, I'} \sum_{j=1}^{n} | \hat{F}_j(I) - \hat{F}_j(I') | \\
&= \max_{I, I'} \sum_{j=1}^{k} | \hat{F}_j(I) - \hat{F}_j(I') | \cdot 1
\end{aligned}
\tag{9}
$$

so that by applying Cauchy-Schwarz inequality, we obtain

$$
\begin{aligned}
\max_{I, I'} \sum_{j=1}^{k} | \hat{F}_j(I) - \hat{F}_j(I') | \cdot 1 &\leq \max_{I, I'} \left( \sum_{j=1}^{k} | \hat{F}_j(I) - \hat{F}_j(I') |^2 \right)^{1/2} \cdot \left( \sum_{j=1}^{k} 1^2 \right)^{1/2} \\
&\leq \max_{I, I'} \| \hat{F}^n(I) - \hat{F}^n(I') \|_2 \cdot \sqrt{k} \\
&\leq \sqrt{k} \cdot \Delta_2(\hat{F}^n).
\end{aligned}
\tag{10}
$$

Consider next (8)(b), which follows since we obtain

$$
\Delta_2(\hat{F}^n) = \max_{I, I'} \| \hat{F}^n(I) - \hat{F}^n(I') \|_2 = \max_{I, I'} \left( \sum_{j=1}^{n} | \hat{F}_j(I) - \hat{F}_j(I') |^2 \right)^{1/2}
\tag{11}
$$

and since $F^n$ has more non-zero elements than $\hat{F}^n$, we have

$$
\Delta_2(\hat{F}^n) \leq \max_{I, I'} \left( \sum_{j=1}^{n} | F_j(I) - F_j(I') |^2 \right)^{1/2}.
\tag{12}
$$

Recall that $F^n(I) = DFT(X^n(I))$, $F^n(I') = DFT(X^n(I'))$, and DFT is linear, so we have

$$
DFT(X^n(I) - X^n(I')) = F^n(I) - F^n(I').
\tag{13}
$$

By applying Parseval's theorem to the DFT, we obtain

$$
\left( \frac{1}{n} \cdot \sum_{j=1}^{n} | F_j(I) - F_j(I') |^2 \right)^{1/2} = \left( \sum_{j=1}^{n} | X_j(I) - X_j(I') |^2 \right)^{1/2}.
\tag{14}
$$

Combining (12) and (14), we prove (8)(b) since we have

$$
\begin{aligned}
\Delta_2(\hat{F}^n) &\leq \max_{I, I'} \sqrt{ \sum_{j=1}^{n} | X_j(I) - X_j(I') |^2 } \cdot \sqrt{n} \\
&\leq \max_{I, I'} \| X^n(I) - X^n(I') \|_2 \cdot \sqrt{n} \\
&\leq \Delta_2(X^n) \cdot \sqrt{n}.
\end{aligned}
\tag{15}
$$

Finally, since the LPA that is applied to $\hat{F}^k$ is $\epsilon$-DP for $\lambda = \frac{\Delta_1(\hat{F}^n)}{\epsilon}$ [22], (8) proves that the FPA is $\epsilon$-DP for $\lambda = \frac{\sqrt{n}\sqrt{k}\Delta_2(X^n)}{\epsilon}$.

**Chunk-based FPA (CFPA).**   One drawback of directly applying the FPA to the eye movement feature signals is large query sensitivities for each feature $f$ due to long signal sizes. To solve this, Steil et al. [20] propose to subsample the signal using non-overlapping windows, which means removing many data points. While subsampling decreases the query sensitivities, it also decreases the amount of data. Instead, we propose to split each signal into smaller

chunks and apply the FPA to each chunk so that complete data can be used. We choose the chunk sizes of 32, 64, and 128 since there are divide-and-conquer type tree-based implementation algorithms for fast DFT calculations when the transform size is a power of 2 [45]. When the signals are split into chunks, chunk level query sensitivities are calculated and used rather than the sensitivity of the whole sequence. Differential privacy for the complete signal is preserved by Theorem 2 [42] since the used chunks are non-overlapping. As the chunk size decreases, the chunk level sensitivity decreases as well as the computational complexity. However, the parameter $\epsilon'$ that accounts for the sample correlations might increase with smaller chunk sizes because temporal correlations between neighboring samples are larger in an eye movement dataset. On the other hand, if the chunk sizes are kept large, then the required amount of noise to achieve differential privacy increases due to the increased query sensitivity. Therefore, a good trade-off between computational complexity, and correlations is needed to determine the optimal chunk size.

**Difference- and chunk-based FPA (DCFPA).** To tackle temporal correlations, we convert the eye movement feature signals into difference signals where differences between consecutive eye movement features are calculated as

$$\hat{X}_t^{(f)} = \{X_t^{(f)} - X_{t-1}^{(f)}\}\big|_{t=2}^n \quad , \quad \hat{X}_1^{(f)} = X_1^{(f)}. \tag{16}$$

Using the difference signals denoted by $\hat{X}^{n,(f)}$, we aim to further decrease the correlations before applying a differential privacy method. We conjecture that the ratio $\epsilon'/\epsilon$ decreases in the difference-based method as compared to the FPA method. To support this conjecture, we show that the correlations in the difference signals decrease significantly as compared to the original signals. This results in lower $\epsilon'$ and better privacy for the same $\epsilon$. The difference-based method is applied together with the CFPA. Therefore, the differences are calculated inside chunks. The first element of each chunk is preserved. Then, the FPA mechanism is applied to the difference signals by using query sensitivities calculated based on differences and chunks. For each chunk, noisy difference observations are aggregated to obtain the final noisy signals. This mechanism is differentially private by Theorem 1 [42], and described in Algorithm 1.

**Algorithm 1:** DCFPA.

**Input:** $X^n$, $\lambda$, $k$

**Outut:** $\tilde{X}^n$

1) $\hat{X}_t = \{X_t - X_{t-1}\}\big|_{t=2}^n \quad , \quad \hat{X}_1 = X_1$.

2) $\tilde{\hat{X}}^n = FPA(\hat{X}^n, \lambda, k)$.

3) $\tilde{X}_t = \{\tilde{\hat{X}}_t + \hat{X}_{t-1}\}\big|_{t=2}^n \quad , \quad \tilde{X}_1 = \tilde{\hat{X}}_1$.

Since Theorem 1 can be applied to the DCFPA when the consecutive differences are assumed to be independent, which is a valid assumption for eye movement feature signals as we illustrate below, there is also a trade-off between the chunk sizes and utility for the DCFPA. If a large chunk size is chosen, then the total $\epsilon$ value could be very large, which reduces privacy. Therefore, we choose chunk sizes of 32, 64, and 128 for the DCFPA as well for evaluation We illustrate the CFPA and DCFPA in Fig 2, for instance with three chunks.

**Choice of the number of transform coefficients.** The proposed methods require a selection of a value for $k$. A small $k$ value increases the reconstruction error, while a large $k$ value results in an increase in the perturbation error. Therefore, it is important to find an optimal $k$ value that minimizes the sum of the two errors. In this work, we compare a large set of possible $k$ values to choose the best values.

We apply the aforementioned differential privacy mechanisms by using 100 noisy evaluations to find optimal $k$ values applied to features or chunks. Optimal $k$ values have the minimum absolute NMSE for each chunk, eye movement feature, and document or recording

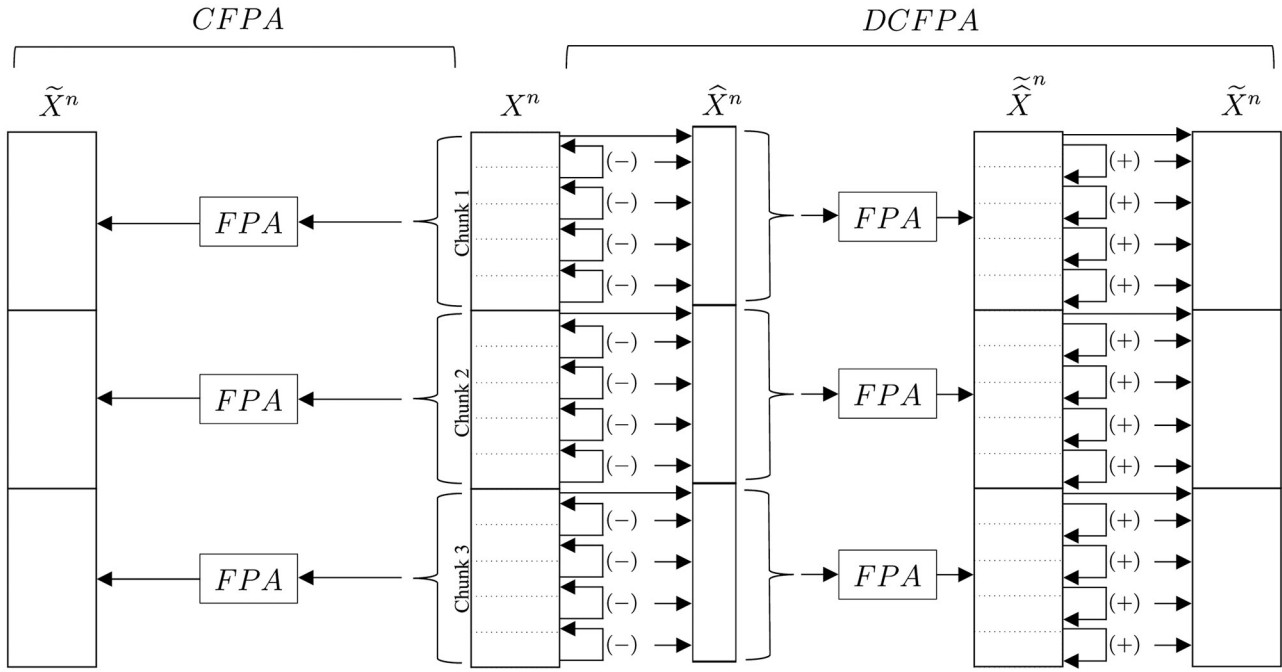

**Fig 2. Flow of the CFPA and DCFPA.**

type. In a distributed setting, each party should know the $k$ values in advance. However, in a centralized setting, it is crucial to choose the $k$ values in a differentially private manner. To evaluate the differential privacy in the eye tracking area while taking the temporal correlations into account, we focus on optimal $k$ values for this work. One shortcoming of this approach is that the optimal $k$ value compromises some information about the data, which leaks privacy [26]. Our observation is that the information leaked by optimizing the parameter $k$ is negligible as compared to the privacy reduction due to temporally correlated data. Thus, we illustrate the results with optimal $k$ values.

## Datasets

We evaluate our methods on two different publicly available eye movement datasets namely, MPIIDPEye and MPIIPrivacEye that are dedicated to privacy-preserving eye tracking. Both datasets consist of aggregated and timely eye movement feature signals related to eye fixations, saccades, blinks, and pupil diameters which are commonly used in VR/AR applications as they represent individual user behaviors. As all minimum values of wordbook features ranging from 1 to 4 are zeros in both datasets, we exclude them from the utility and privacy calculations. In addition, we remark that both datasets are available for non-commercial scientific purposes.

**MPIIDPEye** [20]: A publicly available eye movement dataset consisting of 60 recordings that is collected from VR devices for a reading task of three document types (comics, newspaper, and textbook) from 20 (10 female, 10 male) participants. Each recording consists of 52 eye movement feature sequences computed with a sliding window size of 30 seconds and a step size of 0.5 seconds.

**MPIIPrivacEye** [29]: A publicly available eye movement dataset consisting of 51 recordings from 17 participants for 3 consecutive sessions with a head-mounted eye tracker and a field

camera, which is similar to an AR setup. Each recording consists of 52 eye movement feature sequences computed with a sliding window size of 30 seconds and a step size of 1 second, and each observation is annotated with binary privacy sensitivity levels of the scene that is being viewed. The dataset also consists of scene features extracted with convolutional neural networks. We do not evaluate the last part of the recording 1 of the participant 10, as the eye movement features are not available for this region. To detect the privacy level of the scene that is being viewed, we remark that the scene is very important [46]; however, an individual's eye movements can improve the detection rate when they are fused with the information from the scene.

## Results

This section discusses data correlations in addition to evaluations using utility and classification metrics. The utility and classification results are averaged over 100 noisy evaluations with the optimal $k$ values in MATLAB. We evaluate and compare the utility of differentially private eye movement feature signals by using absolute NMSE, as this metric provides analytically trackable results. However, it does not provide implications regarding the practical usability of the private eye movement signals. Therefore, we also report classification accuracies of document type, scene privacy sensitivity, gender prediction, and person identification tasks in order to show the usability of the private data and proposed methods. An optimal trade-off between utility tasks (e.g., low absolute NMSE, high classification accuracy in document type prediction) and privacy (e.g., low $\epsilon$, low classification accuracy in person identification or gender prediction tasks) is favorable.

### Correlation analysis

Using the correlation coefficient as the metric, we first illustrate high temporal correlation between eye movement feature data. Since there are 52 eye movement features in both datasets, it is not feasible to illustrate all correlation results. Thus, in the following we illustrate the correlations for the features *ratio large saccade* and *blink rate* in the MPIIDPEye and MPIIPrivacEye datasets, respectively. The correlation coefficients of *ratio large saccade* and *blink rate* for three document and recording types over a time difference $\Delta t$ w.r.t. the signal samples at, e.g., the fifth time instance for original eye movement feature signals and difference signals for all participants for both datasets are depicted in Figs 3–6, respectively. As

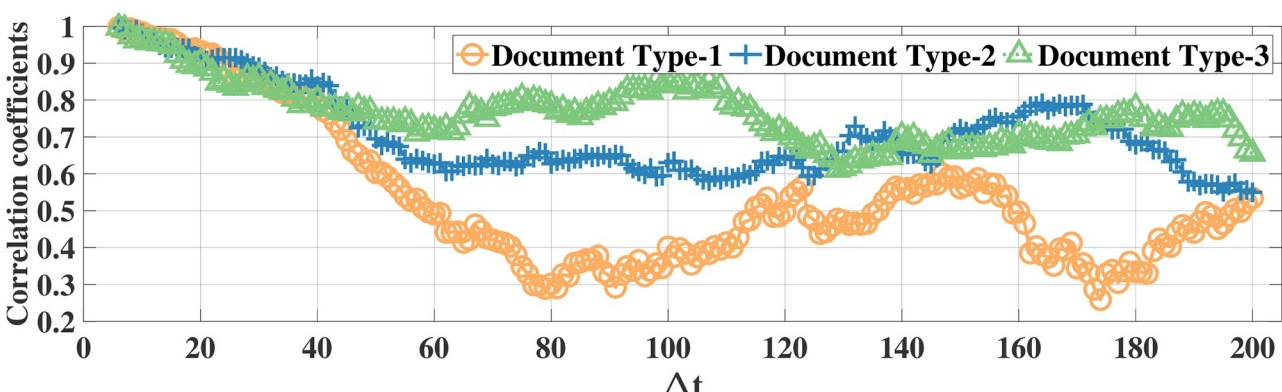

**Fig 3. Correlation coefficients of the raw signals of feature *ratio large saccade* in the MPIIDPEye dataset.** The values are calculated over a time difference of $\Delta t$ (Each time step corresponds to 0.5s) w.r.t. the samples at the fifth time instance.

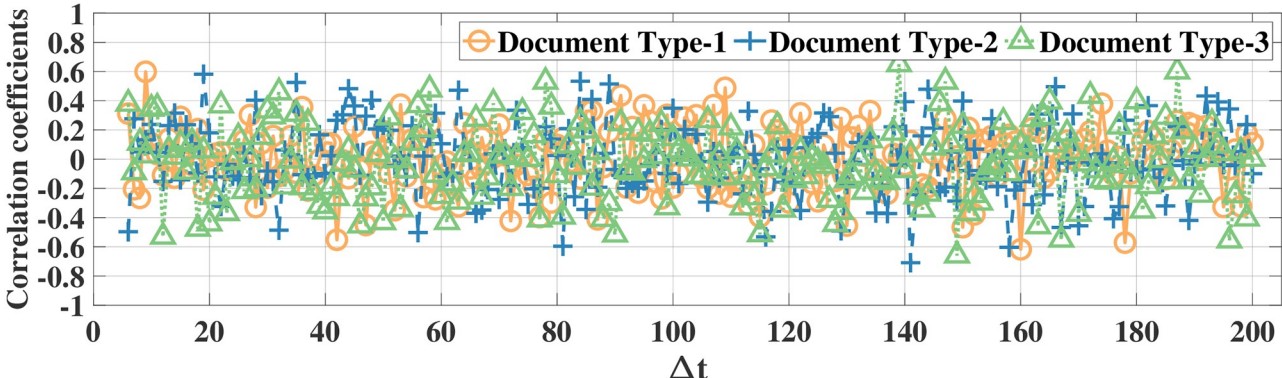

**Fig 4. Correlation coefficients of the difference signals of feature *ratio large saccade* in the MPIIDPEye dataset.** The values are calculated over a time difference of $\Delta t$ (Each time step corresponds to 0.5s) w.r.t. the samples at the fifth time instance.

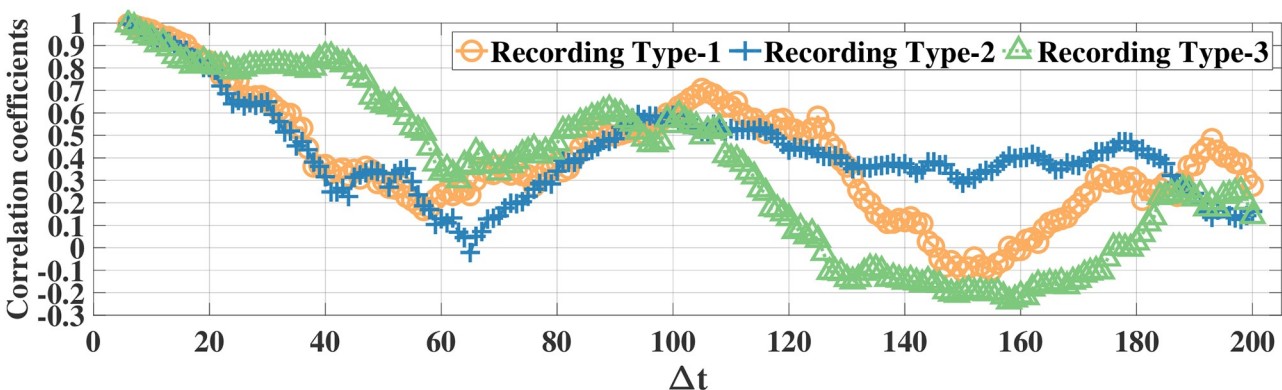

**Fig 5. Correlation coefficients of the raw signals of feature *blink rate* in the MPIIPrivacEye dataset.** The values are calculated over a time difference of $\Delta t$ (Each time step corresponds to 1s) w.r.t. the samples at the fifth time instance.

correlations between the difference signals are significantly smaller than correlations between the original eye movement feature signals, the DCFPA is less vulnerable to privacy reduction due to temporal correlations, thus ensuring that the value of $\epsilon'$ is close to the differential privacy design parameter $\epsilon$.

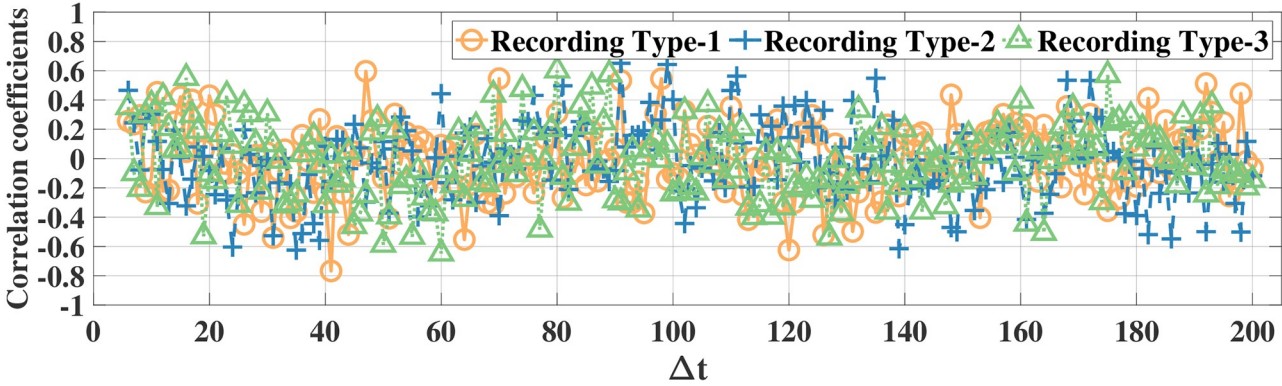

**Fig 6. Correlation coefficients of the difference signals of feature *blink rate* in the MPIIPrivacEye dataset.** The values are calculated over a time difference of $\Delta t$ (Each time step corresponds to 1s) w.r.t. the samples at the fifth time instance.

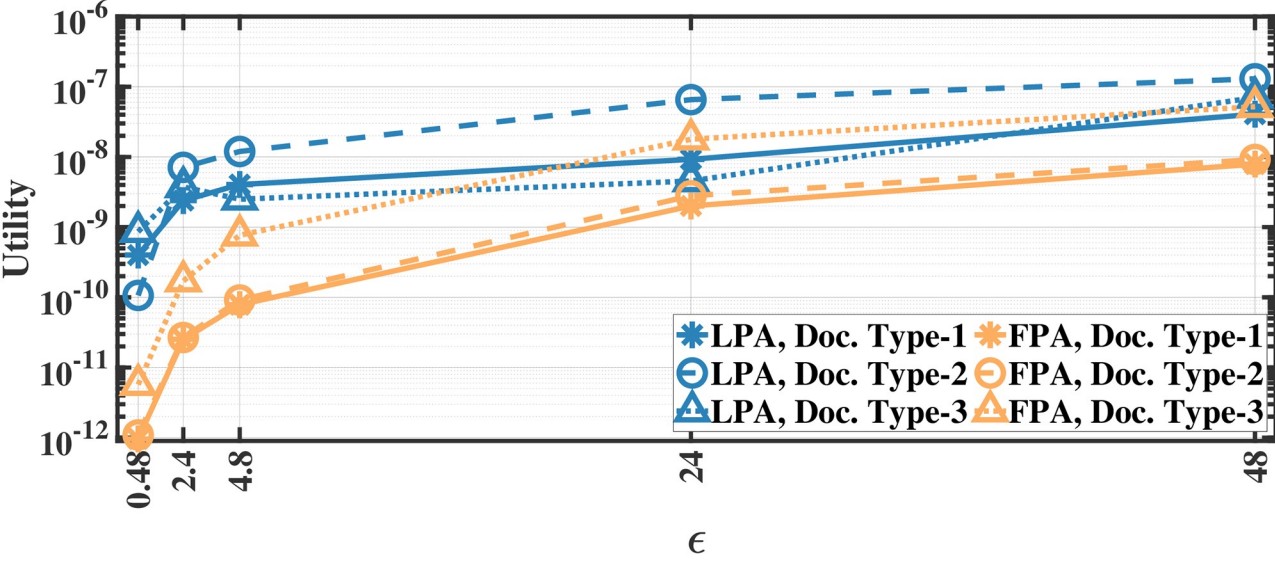

**Fig 7. Utility of the LPA and FPA for MPIIDPEye.**

## Utility results

We evaluate the utility defined in Eq (6) by applying our methods separately to different document and recording types; therefore, we report the utility results separately. As we apply the proposed methods separately to each eye movement feature, we first calculate the mean utility of each feature and then calculate the average utility over all features. The utility results for various $\epsilon$ values for aforementioned methods on the MPIIDPEye and MPIIPrivacEye datasets are given in Figs 7–12, respectively.

While a high absolute NMSE, i.e., low utility, does not necessarily mean that a mechanism is completely useless, higher utility means that the mechanism would perform more effectively

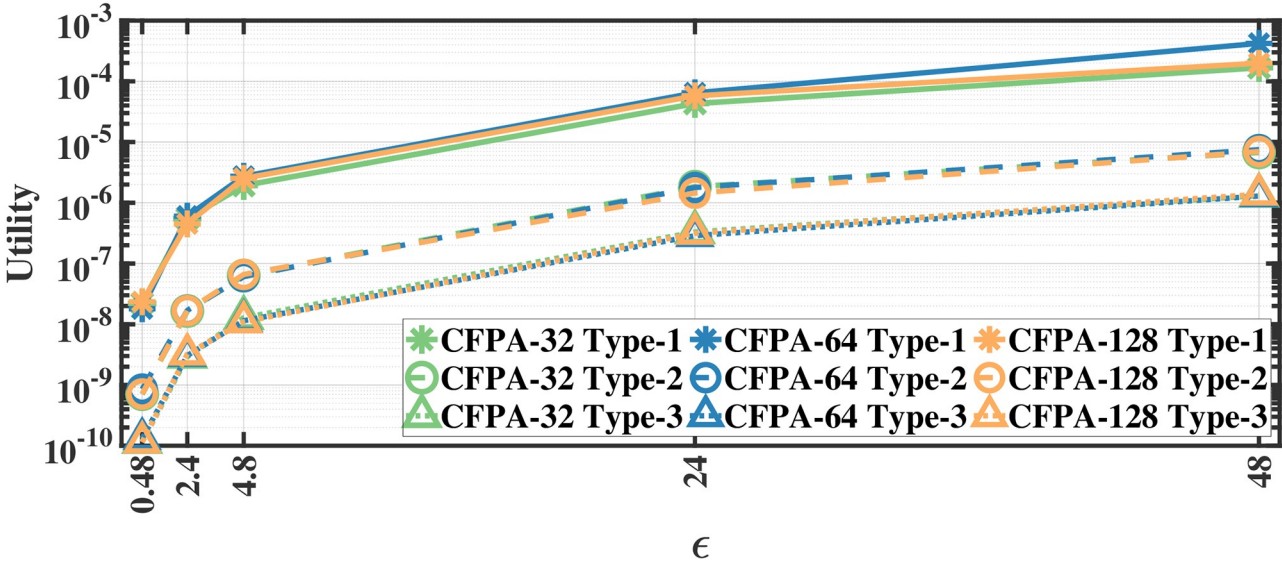

**Fig 8. Utility of the CFPA for MPIIDPEye.**

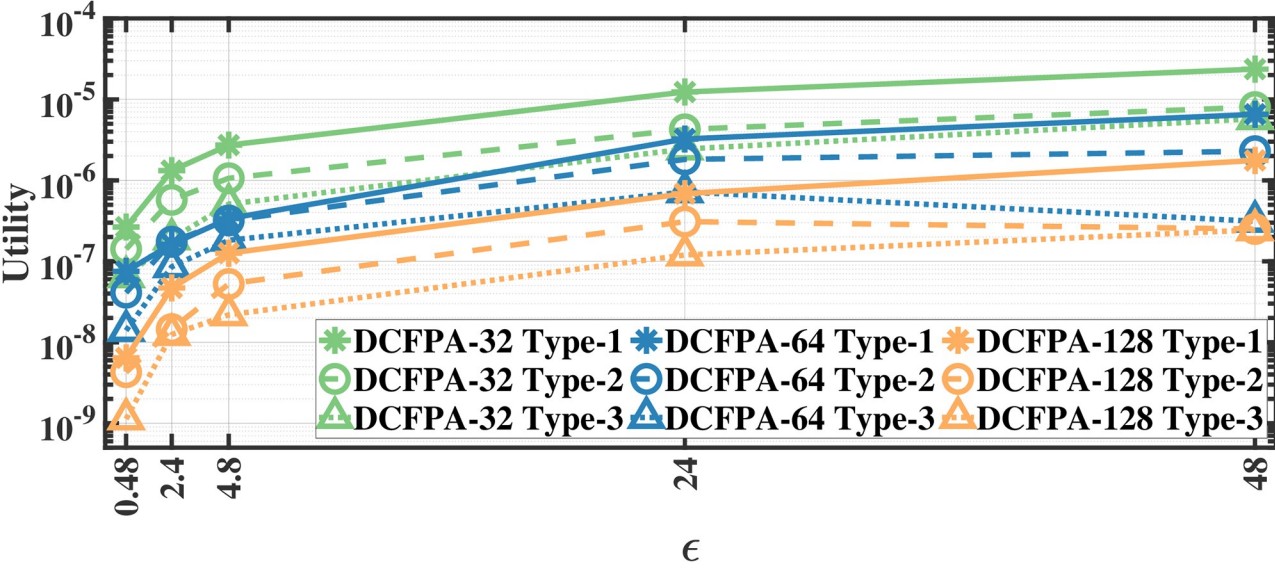

**Fig 9. Utility of the DCFPA for MPIIDPEye.**

than low utility in various tasks. The trend in the utility results of both evaluated datasets are similar. As the query sensitivities are lower in CFPA, utilities of CFPA are always higher than the utilities of the FPA as theoretically expected. The DCFPA particularly with small chunks outperforms other methods in the most private settings, namely in the lowest $\epsilon$ regions. When different chunk sizes are compared within the CFPA and DCFPA, different chunk sizes perform similarly for the CFPA method. For the DCFPA, there is a significant trend for better utilities when the chunk sizes are decreased. However, as temporal correlations in the smaller chunk sizes higher and since a higher chunk size reduces the temporal correlations better, it is ideal to use a higher chunk size if the utilities are comparable. In general, while the LPA,

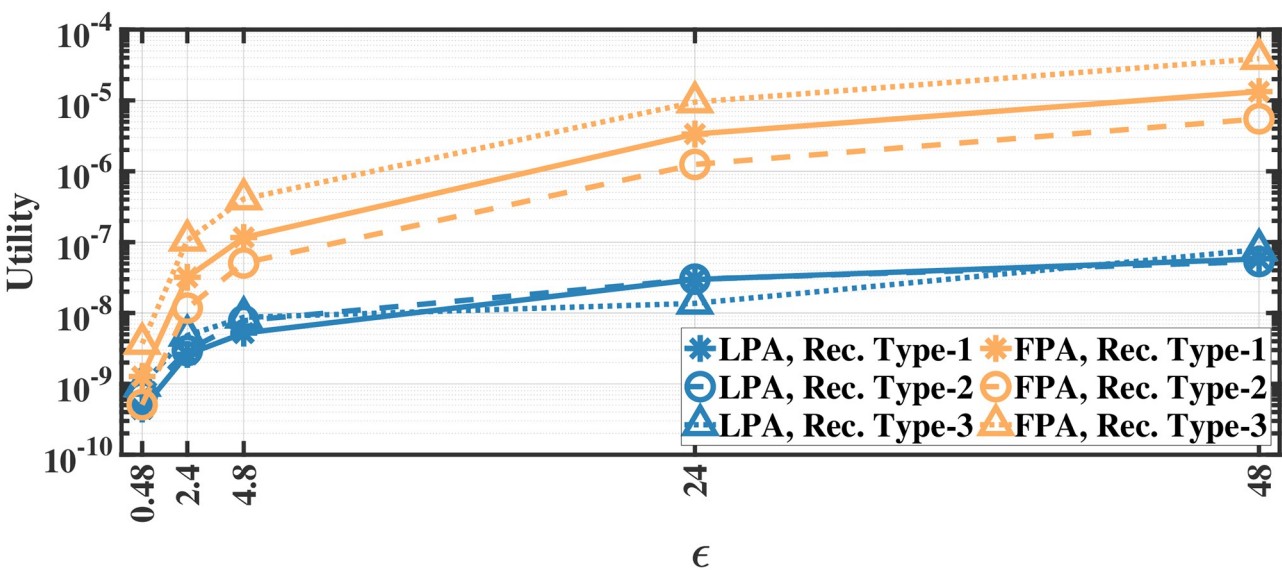

**Fig 10. Utility of the LPA and FPA for MPIIPrivacEye.**

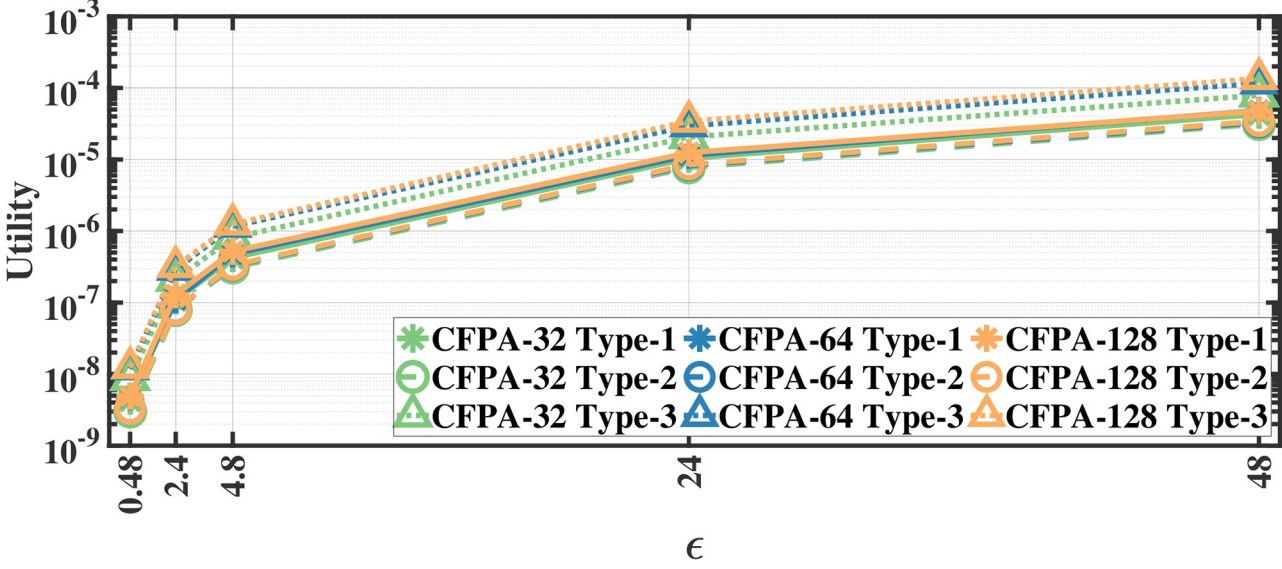

**Fig 11. Utility of the CFPA for MPIIPrivacEye.**

namely the standard Laplace mechanism used for differential privacy, is vulnerable to temporal correlations [41], our methods also outperform it in terms of utilities. In addition to high utilities, the calculation complexities are decreased with the CFPA and DCFPA which is another advantage of chunk-based methods.

## Classification accuracy results

We evaluate document type and gender classification results for the MPIIDPEye and privacy sensitivity classification results for the MPIIPrivacEye by using differentially private data generated by the methods which handle temporal correlations in the privacy context. In addition,

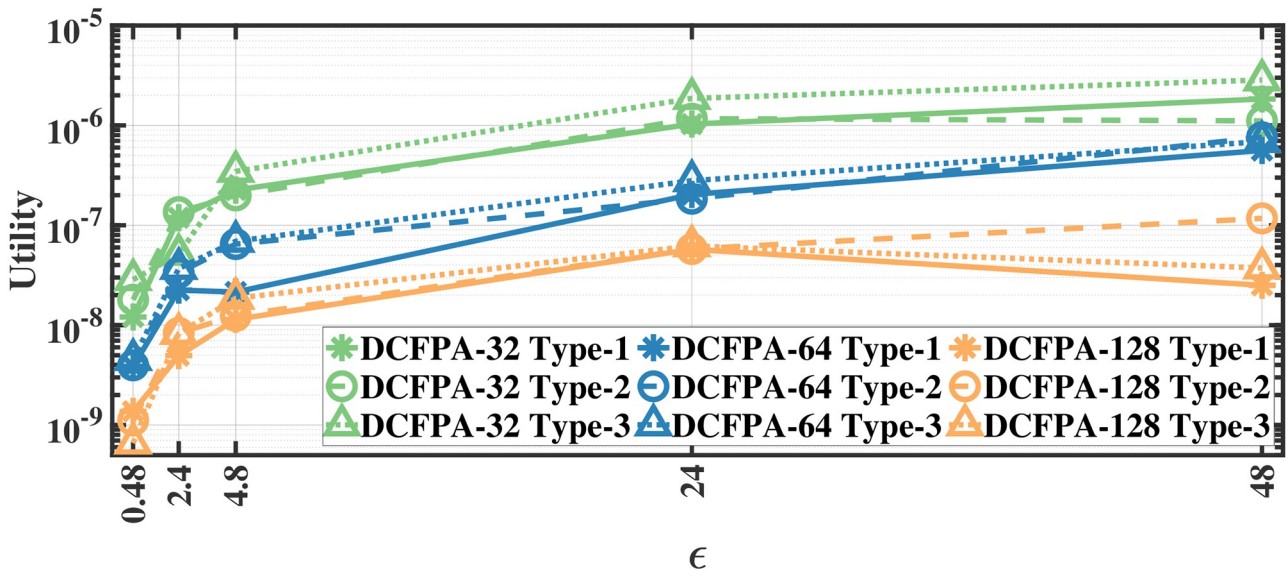

**Fig 12. Utility of the DCFPA for MPIIPrivacEye.**

for both datasets, we evaluate person identification tasks. While a NMSE-based utility metric provides analytically trackable way for comparison, evaluating private data using classification accuracies give insights about the usability of the noisy data in practice. Instead of only using Support Vector Machines (SVM) as in previous works [20, 29], we evaluate a set of classifiers including SVMs, decision trees (DTs), random forests (RFs), and k-Nearest Neighbors (k-NNs). We employ a similar setup as in previous work [20] with radial basis function (RBF) kernel, bias parameter of $C = 1$, and automatic kernel scale for the SVMs. For RFs and k-NNs, we use 10 trees and $k = 11$ with a random tie breaker among tied groups, respectively. We normalize the training data to zero mean and unit variance, and apply the same parameters to the test data. Although we do not apply subsampling while generating the differentially private data, which is applied in previous work [20], we use subsampled data for training and testing for document type, gender, and privacy sensitivity classification tasks with window sizes of 10 and 20 for MPIIDPEye and MPIIPrivacEye, respectively, to have a fair comparison and similar amount of data. Apart from the person identification task, all the classifiers are trained and tested in a leave-one-person-out cross-validation setup, which is considered as a more challenging but generic setup. For the person identification task, since it is not possible to carry out the experiments in a leave-one-person-out cross-validation setup, we opt for a similar configuration as in previous work [20] by using the first halves of the signals as training data and the remaining parts as test data. Such setup can be considered as one of the hypothetical best-case scenarios for an adversary as this simulates some set of prior knowledge for an adversary on participants' visual behaviors. For the person identification task, in order to use similar amount of data with other classification tasks from each signal, we use window sizes of 5 and 10 for MPIIDPEye and MPIIPrivacEye, respectively. For the MPIIDPEye, we evaluate results both with majority voting by summarizing classifications from different time instances for each participant and recording and without majority voting. Privacy sensitivity classification tasks for MPIIPrivacEye are carried only without majority voting since privacy sensitivity of the scene can change at each time step and applying majority voting to such task in our setup is not reasonable.

While classification results cannot be treated directly as the utility, they provide insights into the usability of the differentially private data in practice. We first evaluate document type classification task in the majority voting setting in Table 1 for MPIIDPEye dataset as it is possible to compare our results with the previous work [20]. As previous results quickly drop to the 0.33 guessing probability in high privacy regions, we significantly outperform them particularly with DCFPA and FPA with the accuracies over 0.60 and 0.85, respectively. In the less private regions towards $\epsilon = 48$, this trend still exists with the CFPA and FPA with accuracy results over 0.7 and 0.85. Chunk-based methods perform slightly worse than the FPA in the document

**Table 1. Document type classification accuracies in the MPIIDPEye dataset using differentially private eye movement features with majority voting.**

| Method | Document type classification accuracies (k-NN\|SVM\|DT\|RF) | | | | |
|---|---|---|---|---|---|
| | $\epsilon = 0.48$ | $\epsilon = 2.4$ | $\epsilon = 4.8$ | $\epsilon = 24$ | $\epsilon = 48$ |
| FPA | 0.50\|0.63\|0.82\|**0.87** | 0.51\|0.63\|0.81\|**0.87** | 0.5\|0.61\|0.81\|**0.87** | 0.52\|0.63\|0.82\|**0.87** | 0.52\|0.64\|0.83\|**0.88** |
| CFPA-32 | 0.39\|0.37\|0.45\|0.44 | 0.40\|0.38\|0.45\|0.44 | 0.40\|0.44\|0.46\|0.44 | 0.58\|0.58\|0.55\|0.60 | **0.71**\|0.69\|0.66\|0.66 |
| CFPA-64 | 0.41\|0.36\|0.45\|0.45 | 0.40\|0.37\|0.44\|0.45 | 0.40\|0.41\|0.44\|0.45 | 0.57\|0.59\|0.55\|0.59 | 0.70\|0.70\|0.66\|0.66 |
| CFPA-128 | 0.36\|0.33\|0.45\|0.45 | 0.36\|0.33\|0.44\|0.44 | 0.37\|0.35\|0.44\|0.45 | 0.52\|0.56\|0.52\|0.57 | 0.69\|0.68\|0.64\|0.66 |
| DCFPA-32 | 0.51\|0.37\|0.46\|0.44 | 0.51\|0.36\|0.47\|0.42 | 0.47\|0.35\|0.47\|0.43 | 0.49\|0.37\|0.46\|0.44 | 0.48\|0.36\|0.47\|0.45 |
| DCFPA-64 | 0.61\|0.45\|0.43\|0.41 | 0.55\|0.35\|0.43\|0.41 | 0.56\|0.41\|0.43\|0.41 | **0.60**\|0.43\|0.45\|0.42 | 0.59\|0.40\|0.44\|0.43 |
| DCFPA-128 | **0.64**\|0.45\|0.46\|0.48 | **0.62**\|0.42\|0.45\|0.46 | **0.69**\|0.50\|0.44\|0.46 | 0.57\|0.45\|0.45\|0.46 | 0.60\|0.42\|0.45\|0.46 |

**Table 2. Gender classification accuracies in the MPIIDPEye dataset using differentially private eye movement features with majority voting.**

| | Gender classification accuracies (k-NN\|SVM\|DT\|RF) | | | | |
|---|---|---|---|---|---|
| Method | $\epsilon = 0.48$ | $\epsilon = 2.4$ | $\epsilon = 4.8$ | $\epsilon = 24$ | $\epsilon = 48$ |
| FPA | 0.44\|0.30\|0.43\|0.38 | 0.45\|0.30\|0.41\|0.37 | 0.44\|0.28\|0.41\|0.39 | 0.43\|0.27\|0.43\|0.38 | 0.44\|0.31\|0.42\|0.39 |
| CFPA-32 | 0.04\|0.01\|0.26\|0.24 | 0.05\|0.01\|0.27\|0.25 | 0.05\|0.02\|0.28\|0.27 | 0.36\|0.30\|0.50\|0.45 | 0.62\|0.50\|0.67\|0.53 |
| CFPA-64 | 0.08\|0.05\|0.27\|0.26 | 0.08\|0.04\|0.28\|0.27 | 0.10\|0.06\|0.31\|0.27 | 0.38\|0.34\|0.52\|0.47 | 0.62\|0.51\|0.68\|0.54 |
| CFPA-128 | 0.18\|0.15\|0.32\|0.30 | 0.16\|0.12\|0.31\|0.30 | 0.18\|0.10\|0.32\|0.31 | 0.36\|0.30\|0.50\|0.46 | 0.60\|0.47\|0.68\|0.54 |
| DCFPA-32 | 0.03\|$\approx$0\|0.22\|0.31 | 0.04\|$\approx$0\|0.23\|0.32 | 0.04\|$\approx$0\|0.22\|0.32 | 0.04\|$\approx$0\|0.23\|0.31 | 0.04\|$\approx$0\|0.23\|0.32 |
| DCFPA-64 | 0.04\|$\approx$0\|0.30\|0.33 | 0.04\|$\approx$0\|0.30\|0.34 | 0.04\|$\approx$0\|0.30\|0.32 | 0.04\|$\approx$0\|0.29\|0.34 | 0.03\|$\approx$0\|0.30\|0.34 |
| DCFPA-128 | 0.09\|0.01\|0.34\|0.35 | 0.08\|$\approx$0\|0.32\|0.34 | 0.08\|0.01\|0.32\|0.35 | 0.07\|$\approx$0\|0.33\|0.34 | 0.07\|0.01\|0.34\|0.34 |

type classifications even though the utility of the FPA is lower. We observe that the reading patterns are hidden easier with chunk-based methods; therefore, document type classification task becomes more challenging. This is especially validated with DCFPA methods using different chunk sizes, as DCFPA-128 outperforms smaller chunk-sized DCFPAs even though the sensitivities are higher. Therefore, we conclude that the differential privacy method should be selected for eye movements depending on the further task which will be applied. The document type classification results without majority voting are provided in the table in S1 Table.

Next, we analyze the gender classification results for MPIIDPEye. All methods are able to hide the gender information in the high privacy regions as it is already challenging to identify it with clean data as accuracies are $\approx$0.7 in previous work [20]. While we obtain similar results compared to previous work for the gender classification task, the CFPA method is able to predict gender information correctly in the less private regions, namely $\epsilon = 48$, as it also has the highest utility values in these regions. The FPA applied to the complete signal and the DCFPA are not able to classify genders accurately. We observe that higher amount of noise that is needed by the FPA and noising the fine-grained "difference" information between eye movement observations with DCFPA are the reasons for hiding the gender information successfully in all privacy regions. Overall, the CFPA provides an optimal equilibrium between gender and document type classification success in the less private regions if gender information is not considered as a feature that should be protected from adversaries. Otherwise, all proposed methods are able to hide gender information from the data in the higher privacy regions as expected. Gender classification results are depicted in Table 2. Especially in some methods with k-NNs and SVMs, gender classification accuracies are close to zero because of the majority voting and it is validated by the results without majority voting in the table in S2 Table.

Lastly for the MPIIDPEye, we evaluate person identification task using differentially private data. The resulting classification accuracies with majority voting are depicted in Table 3. By

**Table 3. Person identification accuracies in the MPIIDPEye dataset using differentially private eye movement features with majority voting.**

| | Person identification accuracies (k-NN\|SVM\|DT\|RF) | | | | |
|---|---|---|---|---|---|
| Method | $\epsilon = 0.48$ | $\epsilon = 2.4$ | $\epsilon = 4.8$ | $\epsilon = 24$ | $\epsilon = 48$ |
| FPA | 1 | 1 | 1 | 1 | 1 |
| CFPA-32 | 0.15\|0.08\|0.44\|0.37 | 0.13\|0.08\|0.46\|0.39 | 0.11\|0.08\|0.48\|0.41 | 0.40\|0.11\|0.64\|0.70 | 0.72\|0.16\|0.87\|0.92 |
| CFPA-64 | 0.14\|0.08\|0.42\|0.34 | 0.13\|0.08\|0.44\|0.37 | 0.12\|0.08\|0.45\|0.38 | 0.39\|0.11\|0.63\|0.71 | 0.70\|0.17\|0.85\|0.92 |
| CFPA-128 | 0.16\|0.05\|0.39\|0.36 | 0.15\|0.05\|0.41\|0.36 | 0.17\|0.05\|0.43\|0.39 | 0.45\|0.07\|0.55\|0.63 | 0.70\|0.16\|0.74\|0.88 |
| DCFPA-32 | 0.06\|0.10\|0.39\|0.37 | 0.06\|0.10\|0.39\|0.36 | 0.08\|0.10\|0.39\|0.36 | 0.10\|0.10\|0.39\|0.37 | 0.10\|0.10\|0.40\|0.38 |
| DCFPA-64 | 0.10\|0.10\|0.33\|0.35 | 0.10\|0.10\|0.32\|0.34 | 0.10\|0.10\|0.32\|0.33 | 0.13\|0.10\|0.31\|0.34 | 0.13\|0.10\|0.32\|0.33 |
| DCFPA-128 | 0.09\|0.05\|0.24\|0.28 | 0.09\|0.05\|0.25\|0.27 | 0.10\|0.05\|0.23\|0.27 | 0.10\|0.06\|0.24\|0.26 | 0.10\|0.05\|0.22\|0.25 |

**Table 4. Privacy sensitivity classification accuracies in the MPIIPrivacEye dataset using differentially private eye movement features.**

| Method | Privacy sensitivity classification accuracies (k-NN\|SVM\|DT\|RF) | | | | |
|---|---|---|---|---|---|
| | $\epsilon = 0.48$ | $\epsilon = 2.4$ | $\epsilon = 4.8$ | $\epsilon = 24$ | $\epsilon = 48$ |
| FPA | 0.49\|0.58\|0.51\|0.55 | 0.49\|0.58\|0.51\|0.55 | 0.49\|0.58\|0.51\|0.55 | 0.50\|0.58\|0.51\|0.55 | 0.50\|0.59\|0.51\|0.55 |
| CFPA-32 | 0.55\|0.59\|0.52\|0.56 | 0.55\|0.58\|0.52\|0.56 | 0.55\|0.58\|0.52\|0.56 | 0.56\|0.58\|0.53\|0.57 | 0.58\|**0.60**\|0.54\|0.58 |
| CFPA-64 | 0.55\|0.58\|0.52\|0.56 | 0.55\|0.58\|0.52\|0.56 | 0.55\|0.58\|0.52\|0.56 | 0.56\|0.58\|0.53\|0.57 | 0.58\|0.59\|0.54\|0.58 |
| CFPA-128 | 0.55\|0.57\|0.52\|0.56 | 0.55\|0.57\|0.52\|0.56 | 0.55\|0.57\|0.52\|0.56 | 0.56\|0.58\|0.53\|0.57 | 0.58\|0.59\|0.54\|0.59 |
| DCFPA-32 | 0.54\|**0.59**\|0.52\|0.56 | 0.55\|**0.59**\|0.52\|0.56 | 0.55\|**0.59**\|0.52\|0.56 | 0.54\|**0.59**\|0.52\|0.56 | 0.55\|0.59\|0.52\|0.56 |
| DCFPA-64 | 0.54\|0.58\|0.52\|0.56 | 0.54\|0.58\|0.52\|0.56 | 0.54\|0.58\|0.52\|0.56 | 0.54\|0.58\|0.52\|0.56 | 0.54\|0.58\|0.52\|0.56 |
| DCFPA-128 | 0.54\|0.57\|0.52\|0.56 | 0.54\|0.57\|0.52\|0.56 | 0.54\|0.57\|0.52\|0.56 | 0.54\|0.57\|0.52\|0.56 | 0.54\|0.57\|0.52\|0.56 |

using the FPA, it is possible to identify the participants very accurately, which means that even though the document type classification accuracies of the FPA are higher than the others, a strong adversary can also identify personal ids when this method is used. The same trend also exists in the without majority voting setting, which is reported in the table in S3 Table. The CFPA and DCFPA perform well against person identification attempts in the high privacy regions. However, when the CFPA is used, it is possible to identify personal ids in the less private regions. Overall, for the MPIIDPEye dataset, the DCFPA performs better than the others due to its resistance against person identification and gender classification and relatively high classification accuracies for the document type predictions. We conclude that this is due to the robust decorrelation effect of the DCFPA.

For the MPIIPrivacEye, we report privacy sensitivity classification accuracies using differentially private eye movement features in the Table 4. The FPA performs worse than our methods. The DCFPA, particularly with the chunk size of 32, outperforms all other methods slightly in the higher privacy regions as it is also the case for the utility results. In the lower privacy regions, the CFPA performs the best with ≈0.60 accuracy. Since performance does not drop significantly in the higher chunk sizes, it is reasonable to use higher chunk-sized methods as they decrease the temporal correlations better. While having an accuracy of approximately 0.6 in a binary classification problem does not form the best performance, according to the previous work [29], privacy sensitivity classification using only eye movements with clean data in a person-independent setup only performs marginally higher than 0.60. Therefore, we show that even though we use differentially private data in the most private settings, we obtain similar results to the classification results using clean data. This means that differentially private eye movements can be used along with scene features for detecting privacy sensitive scenes in AR setups.

The results of the person identification task in the MPIIPrivacEye dataset are similar to the results of the MPIIDPEye dataset and the results with majority voting are depicted in Table 5. Personal identifiers are predicted very accurately when the FPA is used. The CFPA and DCFPA are resistant to person identification attacks in all privacy regions performing around the random guess probability in almost all cases. Similar to the classification results of the MPIIDPEye dataset, the DCFPA method performs the best when utility-privacy trade-off is taken into consideration. The person identification results without majority voting are presented in the table in S4 Table.

## Discussion

We compared our differential privacy methods with the standard Laplace mechanism as well as the FPA method, which is proposed for temporally correlated data, by using the MPIIDPEye

**Table 5. Person identification classification accuracies in the MPIIPrivacEye dataset using differentially private eye movement features with majority voting.**

| | Person identification classification accuracies (k-NN\|SVM\|DT\|RF) | | | | |
|---|---|---|---|---|---|
| Method | $\epsilon = 0.48$ | $\epsilon = 2.4$ | $\epsilon = 4.8$ | $\epsilon = 24$ | $\epsilon = 48$ |
| FPA | 1 | 1 | 1 | 1 | 1 |
| CFPA-32 | 0.05\|0.06\|0.07\|0.07 | 0.05\|0.06\|0.07\|0.07 | 0.06\|0.06\|0.08\|0.07 | 0.07\|0.06\|0.09\|0.11 | 0.11\|0.06\|0.14\|0.16 |
| CFPA-64 | 0.06\|0.06\|0.06\|0.07 | 0.06\|0.06\|0.06\|0.06 | 0.06\|0.06\|0.07\|0.07 | 0.07\|0.06\|0.09\|0.09 | 0.11\|0.06\|0.16\|0.16 |
| CFPA-128 | 0.06\|0.06\|0.07\|0.07 | 0.06\|0.06\|0.07\|0.07 | 0.06\|0.06\|0.07\|0.08 | 0.07\|0.06\|0.09\|0.11 | 0.11\|0.06\|0.15\|0.15 |
| DCFPA-32 | 0.06\|0.05\|0.08\|0.07 | 0.06\|0.06\|0.07\|0.08 | 0.07\|0.05\|0.08\|0.08 | 0.07\|0.05\|0.08\|0.08 | 0.07\|0.06\|0.08\|0.08 |
| DCFPA-64 | 0.06\|0.05\|0.06\|0.06 | 0.06\|0.05\|0.06\|0.06 | 0.06\|0.05\|0.06\|0.06 | 0.05\|0.06\|0.06\|0.06 | 0.06\|0.05\|0.06\|0.06 |
| DCFPA-128 | 0.05\|0.05\|0.06\|0.06 | 0.05\|0.05\|0.05\|0.06 | 0.06\|0.05\|0.06\|0.06 | 0.05\|0.05\|0.05\|0.06 | 0.06\|0.05\|0.05\|0.06 |

and MPIIPrivacEye datasets. The utility results based on the NMSE metric show that due to the reduced sensitivities as a result of the chunking operations, the CFPA and DCFPA perform better than the FPA and standard Laplace mechanism. While larger chunk sizes applied with the CFPA and DCFPA decrease the effects of temporal correlations on the differential privacy mechanisms, they also increase the sensitivities, leading to higher amount of noise addition to the data and worse utility performance. Utility evaluations represent how much differentially private signals diverge from the original signals. Having eye movement feature signals less diverged from the original values by providing the differential privacy would lead better performance in various tasks. While the FPA, CFPA, and DCFPA are appropriate for temporally correlated data, the DCFPA uses the consecutive differences of eye movement feature signals, which are significantly less correlated than the original feature signals, as illustrated in Figs 4 and 6. Thus, the DCFPA is less vulnerable to temporal correlations in the differential privacy context.

In addition to utility results, we evaluated document type, gender, and person identification tasks for the MPIIDPEye dataset and privacy sensitivity classification of the observed scene and person identification task for the MPIIPrivacEye dataset and compared our results with the previous works especially in the eye tracking literature. The FPA outperforms the CFPA and DCFPA in document type classification task since the chunks perturb "Z"-type reading patterns. However, this might be a task-specific outcome as the CFPA and DCFPA perform better in terms of utility. In addition, when the FPA is used, personal identifiers can be estimated with high accuracies in both datasets. On the contrary, especially the DCFPA provides decreased probabilities for person identification tasks in the MPIIDPEye, and probabilities close to the random guess probability for the MPIIPrivacEye, which are optimal from a privacy-preservation perspective. We remark that this outcome is also related to decreased temporal correlations. Gender information is successfully hidden in all methods and scene privacy can be predicted to some extent using differentially private eye movement signals. In addition, privacy sensitivity detection results on the MPIIPrivacEye are consistent with the utility results based on the NMSE metric.

Due to the significant reduction of temporal correlations and high utility and relatively accurate classification results in different tasks, the DCFPA is the best performing differential privacy method for eye movement feature signals. In addition, it is not possible to recognize the person accurately from eye movement data when the DCFPA is used. From correlation reduction point of view, in both methods namely, CFPA and DCFPA, when the performances are similar, it is reasonable to use higher chunk sizes as such chunks are less vulnerable to temporal correlations as illustrated in Figs 3 and 5. Overall, our methods outperform the state-of-the-art for differential privacy for aggregated eye movement feature signals.

## Conclusion

We proposed different methods to achieve differential privacy for eye movement feature signals by correcting, extending, and adapting the FPA method. Since eye movement features are correlated over time and are high dimensional, standard differential privacy methods provide low utility and are vulnerable to inference attacks. Thus, we proposed privacy solutions for temporally correlated eye movement data. Our methods can be easily applied to other biometric human-computer interaction data as well since they are independent of the used data and outperform the state-of-the-art methods in terms of both NMSE and classification accuracy and reduce the correlations significantly. In future work, we will analyze the actual privacy metric $\epsilon'$ which takes the data correlations into account and choose $k$ values in a private manner for the centralized differential privacy setting.

## Supporting information

**S1 Table. Document type classification results without majority voting for the MPIIDPEye dataset.**
(PDF)

**S2 Table. Gender classification results without majority voting for the MPIIDPEye dataset.**
(PDF)

**S3 Table. Person identification results without majority voting for the MPIIDPEye dataset.**
(PDF)

**S4 Table. Person identification results without majority voting for MPIIPrivacEye dataset.**
(PDF)

## Acknowledgments

O. Günlü thanks Ravi Tandon for his useful suggestions. E. Bozkir thanks Martin Pawelczyk and Mete Akgün for useful discussions.

## Author Contributions

**Conceptualization:** Efe Bozkir, Onur Günlü, Wolfgang Fuhl, Rafael F. Schaefer, Enkelejda Kasneci.

**Data curation:** Efe Bozkir, Onur Günlü.

**Formal analysis:** Efe Bozkir, Onur Günlü.

**Investigation:** Efe Bozkir, Onur Günlü.

**Methodology:** Efe Bozkir, Onur Günlü.

**Software:** Efe Bozkir, Onur Günlü.

**Supervision:** Efe Bozkir, Onur Günlü, Wolfgang Fuhl, Rafael F. Schaefer, Enkelejda Kasneci.

**Validation:** Efe Bozkir, Onur Günlü.

**Visualization:** Efe Bozkir, Onur Günlü.

**Writing – original draft:** Efe Bozkir, Onur Günlü.

**Writing – review & editing:** Efe Bozkir, Onur Günlü, Wolfgang Fuhl, Rafael F. Schaefer, Enkelejda Kasneci.

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
