## [Decision Letter · Decision Letter 0]

16 Apr 2021

PONE-D-21-07774

Differential privacy for eye tracking with temporal correlations

PLOS ONE

Dear Dr. Bozkir,

Thank you for submitting your manuscript to PLOS ONE. After careful consideration, we feel that it has merit but does not fully meet PLOS ONE’s publication criteria as it currently stands. Therefore, we invite you to submit a revised version of the manuscript that addresses the points raised during the review process.

In particular:

Make the motivation of this work clearer (why eye movement data are sensitive? why DP is vulnerable under temporal correlation? why it is reasonable to assume a trusted central server in this work?).Please try to provide formal proofs of DP achieved by the proposed method. While this paper spans both biomedical engineering and security, most readers in the security domain would expect that claims about privacy are backed by formal proofs.Consider adding related works as suggested by the reviewers.Consider testing the methods on other datasets or provide arguments why this is not possible / not required.Provide a comparison with existing methods in the literature, especially in terms of privacy rather than reconstruction of noisy data and its classification (or provide arguments why the latter covers the former).Remove / reduce parts that appear to be disconnected form the rest of the paper or make the link clearer and justified.

We look forward to receiving your revised manuscript.

Kind regards,

Luca Citi, PhD

Academic Editor

PLOS ONE

Journal Requirements:

Reviewers' comments:

Reviewer's Responses to Questions

**Comments to the Author**

1. Is the manuscript technically sound, and do the data support the conclusions?

Reviewer #1: Yes

Reviewer #2: No

Reviewer #3: Partly

2. Has the statistical analysis been performed appropriately and rigorously? 

Reviewer #1: Yes

Reviewer #2: No

Reviewer #3: Yes

3. Have the authors made all data underlying the findings in their manuscript fully available?

Reviewer #1: Yes

Reviewer #2: No

Reviewer #3: No

4. Is the manuscript presented in an intelligible fashion and written in standard English?

Reviewer #1: Yes

Reviewer #2: Yes

Reviewer #3: Yes

5. Review Comments to the Author

Reviewer #1: The paper “Differential privacy for eye tracking with temporal correlations” has proposed a novel data encoding technique based on differential privacy mechanism to preserve privacy of eye data for various VR/AR applications taking into account temporal correlation.

Specifically, the authors have extended the standard Laplacian Perturbation Algorithm (LPA) and Fourier Perturbation Algorithm (FPA) and have applied them for noise generation and dealing with the eye data correlation issue respectively.

Results and tables presented have significantly shown and proven chunk-based FPA (CFPA) and Difference- and chunk-based FPA (DCFPA), both as extended FPA proposed by authors are more efficient for privacy (data utilities) but provides low usability (data identification or classification) as compared to the standard FPA thus validating author’s claims on proposed methods and conclusions.

The paper is well written and clearly explained. Notwithstanding authors may consider the following minor suggestions:

• Spotted typo in abstract, “extend” instead of “extent”

• To further demonstrate the versatility of the proposed method, authors may consider expanding their evaluation to include other configurations that are not limited to MPIIDPEye and MPIIPrivacEye datasets. Alternatively, authors may give more justification to why proposed method evaluation was limited to MPIIDPEye and MPIIPrivacEye configurations other than just being the only non-commercial scientific purpose datasets.

Reviewer #2: The authors are studying an interesting topic of privacy-preserving eye tracking data release, which is important given the rapid advancement of VR and AR technologies. However, this paper has a lot room to improve. Here I list several essential weak points in the hope that the authors could provide a better version of this work.

W1. The motivation of this work needs to be clarified. For example, why eye movement data are sensitive? Why DP is vulnerable under temporal correlation? (also see W3) Why it is reasonable to assume a trusted central server in this work? (basically, LDP is perforable than central DP since LDP has less trust assumption)

W2. The technical depth of the proposal method is limited. The proposed Chunk-based and difference-based methods seem trivial to me. The privacy guarantee is not clear; especially, how they could guarantee DP under temporal correlations. The authors may want to provide the formal proofs.

W3. Important related works are missing. The following studies [a,b] demonstrate the vulnerability of DP under temporal correlations. The authors fail to acknowledge them and did not discuss whether or not the proposed methods in this work can address the vulnerability proven in [a,b].

[a] Quantifying Differential Privacy under Temporal Correlations, IEEE ICDE 2017.

[b] Quantifying Differential Privacy in Continuous Data Release Under Temporal Correlations. IEEE TKDE 2019.

Reviewer #3: Bozkir et al proposed a differential privacy method based on temporal signal processing methods to overcome possibly biomimetic information eavesdropping or information stealing. Their results focus on the correlation analysis of public available data in order to show different possible signal states. Then they apply a utility function in order to analyse how different techniques extended from the temporal signal processing methods will lead to less error in
classifying the signal states when noise is added to the raw signal, thus implementing the differential privacy scheme.

The work is interesting and certainly has a timely contribution to this area. However, from the paper's current state, it is not clear weather there is significant benefits to privacy schemes of eye movement data because i) the authors do not provide a direct comparison with existing methods in the literature, and most of the evaluated techniques are based on simple extension of temporal signal processing methods, ii) the metrics used for evaluating the privacy technique are focused
on reconstruction of noisy data and its classification, but not entirely on how efficient in terms of privacy the proposed techniques really are.

Moreover, in the last part of the paper, the authors spend a considerable amount of space on a totally different focus, which is classifying the different cohort features in the obtained datasets. This part has not a clear link to the differential privacy techniques and thus takes away a lot from the main message of the paper. I recommend the authors would make more space for evaluating further the proposed privacy methods directly.

However, I do think if the authors address the main points, this paper could lead to nice findings on how evaluating temporal signals correctly address the challenges in differential privacy.

Minor comments below:

- In: "Soon, the decrease in the cost of such devices might cause a mass consumption across different application domains such as gaming, entertainment, or education. " - How soon and a decrease from how much? Consider more precision in such statements.

- In: "As biometric contents can be retrieved from eye movements, it is important to protect them against adversarial attacks." - At this point the reader may not necessarily know what an adversarial attack is. Therefore, add a sentence here about what are adversarial attacks and why they are important.

- In: "To apply differential privacy to the eye movement data, we evaluate the standard Laplacian Perturbation Algorithm (LPA) [22] and Fourier Perturbation Algorithm (FPA) [25]. " - Why chose particularly those two algorithms? This must be motivated.

- In: "sing differential privacy, noise is added to the outcome of a function so that the outcome does not significantly change based on whether or not a randomly chosen individual participated in the dataset." - This sentence is confusing.

- In: "In addition, it is not possible to recognize the person accurately from eye movement data when the DCFPA is used." - Here the authors also show how the classification work, in the last part of the paper, may also not be a good result, which reinforces the need to either shrink it down or remove it completely.

- In: "Our methods and the current results form the state-of-the-art for differential privacy for aggregated eye movement feature signals." - This sentence is incomplete.

6. PLOS authors have the option to publish the peer review history of their article (what does this mean?). If published, this will include your full peer review and any attached files.

Reviewer #1: No

Reviewer #2: No

Reviewer #3: No

---

## [Author Response · Author response to Decision Letter 0]

27 May 2021

We thank the Academic Editor and the Reviewers for their useful suggestions and comments that significantly helped to improve the paper, and provide our rebuttal letter below, following the order of reviewers’ comments. The main comments of the Academic Editor and the Reviewers are enumerated, and our responses are provided below the enumerated parts. In addition, we improved the readability of our manuscript. Please also see the revised version of our manuscript for the changes
made. We hope that all concerns have been addressed in a satisfying way.

Response to the Academic Editor:

1) Make the motivation of this work clearer (why eye movement data are sensitive? why DP is vulnerable under temporal correlation? why it is reasonable to assume a trusted central server in this work?).

Thank you very much for your request to clarify our motivations. Since it is possible to authenticate and identify individuals by using solely their eye movements data, they are considered as sensitive information. We motivate this better in the revised version of the “Introduction” section of our paper (Please refer to the first paragraph of the “Introduction”.). We also explain the vulnerability of the DP to the temporal correlations and the assumption of
the trusted server in our rebuttal below under the points (10) and (11), respectively. We especially motivate the temporal correlation issue in our paper now much clearly also by including the works suggested by the Reviewer 2 (See the “Previous research” section of our revised manuscript.).

2) Please try to provide formal proofs of DP achieved by the proposed method. While this paper spans both biomedical engineering and security, most readers in the security domain would expect that claims about privacy are backed by formal proofs.

In the revised manuscript, we provide the formal proof of FPA, which is our proof and which corrects the mistaken proof in the literature, and discuss the reasons why the chunk- and difference-based methods preserve the privacy via parallel and sequential compositions in CFPA and DCFPA sections, respectively. 

3) Consider adding related works as suggested by the reviewers.

We have added the related work suggested by the reviewers and discussed them along with our methods especially in the “Previous research” section. Please see point (13) below for more details.

4) Consider testing the methods on other datasets or provide arguments why this is not possible / not required.

We have evaluated our methods on two prominent, recent, and publicly available eye movement datasets that are obtained from VR/AR setups. As the decorrelation effect that is provided by the Fourier transformation, partitioning into chunks, and converting the data to difference (or subtracted) signals does not strongly depend on the datasets (due to the strong decorrelation efficiency of the Fourier transform), it is reasonable to say that testing the proposed methods on other datasets
is not necessary from a theoretical perspective. For example, by partitioning the data into chunks, the query sensitivities that should be applied to achieve the differential privacy decrease automatically and it is valid for any data (i.e., for any probabilistic model for the data). One trade-off that might depend on the datasets is the required sizes of used chunks to achieve a target level of decorrelation effect. We have already discussed this fact in the “Methods”
section of our paper. If very small sized chunks are used, the data might still be temporally correlated, which increases the actual privacy metric epsilon-prime. With larger chunk sizes, temporal correlations between the chunks decrease; however, then the query sensitivities increase. Therefore, when applying the proposed methods, this trade-off should be taken into consideration to decide a reasonable chunk size for a given dataset. This is essentially not directly related to the raw
eye movement data, but to the feature extraction pipeline of aggregated eye movement features. For instance, in the MPIIDPEye and MPIIPrivacEye datasets, the sliding window size was selected as 0.5 and 1 seconds during the feature generation, respectively. If a larger sliding window is selected, the temporal correlations will be lower in the aggregated features. This shows that an empirical evaluation and decision should be made when selecting the chunk sizes to generate private data.
Other than these points, the proposed methods are generic and could be applied to all datasets or data types. We motivate this insight in a better way in the updated manuscript.

5) Provide a comparison with existing methods in the literature, especially in terms of privacy rather than reconstruction of noisy data and its classification (or provide arguments why the latter covers the former).

We have already compared our results with the existing methods that apply differential privacy to the aggregated eye movement features, the standard Laplacian mechanism used to provide differential privacy, and the (corrected) FPA method that is suitable for time-series data, and we used the NMSE and classification accuracy metrics as our utility metric for comparisons. We clarify these below under point (14) in our rebuttal and strengthen our argumentations in our revised version of
the manuscript. 

6) Remove / reduce parts that appear to be disconnected form the rest of the paper or make the link clearer and justified.

Thank you very much for your suggestion! We have justified the reasons for using classification accuracy metric in our paper at the beginning of the “Results” section and under the point (16) below. We argue that the normalized mean square error (NMSE) metric is an analytically trackable utility metric to compare the original and private versions of the signals, whereas classification accuracies of different tasks (which is another utility metric that is difficult to track
analytically) provides insights into the practical usability of the private eye movement signals. Such an evaluation scheme is also in line with the previous research in the field of privacy-preserving eye tracking (Steil et al. 2019 [20], David-John et al. 2021 [34] in the paper), as the main goal of these works is to make the persons unidentifiable while keeping the utility of the data as high as possible.

Thank you very much for your comments!

Response to the Reviewer #1:

Reviewer #1: The paper “Differential privacy for eye tracking with temporal correlations” has proposed a novel data encoding technique based on differential privacy mechanism to preserve privacy of eye data for various VR/AR applications taking into account temporal correlation. Specifically, the authors have extended the standard Laplacian Perturbation Algorithm (LPA) and Fourier Perturbation Algorithm (FPA) and have applied them for noise generation and dealing with the
eye data correlation issue respectively.

Results and tables presented have significantly shown and proven chunk-based FPA (CFPA) and Difference- and chunk-based FPA (DCFPA), both as extended FPA proposed by authors are more efficient for privacy (data utilities) but provides low usability (data identification or classification) as compared to the standard FPA thus validating author’s claims on proposed methods and conclusions. The paper is well written and clearly explained. Notwithstanding authors may consider the
following minor suggestions:

7) Spotted typo in abstract, “extend” instead of “extent”

We have updated the typo in the abstract from “extent” to “extend”. Thank you!

8) To further demonstrate the versatility of the proposed method, authors may consider expanding their evaluation to include other configurations that are not limited to MPIIDPEye and MPIIPrivacEye datasets. Alternatively, authors may give more justification to why proposed method evaluation was limited to MPIIDPEye and MPIIPrivacEye configurations other than just being the only non-commercial scientific purpose datasets.

We use the MPIIDPEye and MPIIPrivacEye datasets since these datasets are benchmarks dedicated to privacy-preserving eye tracking in the VR/AR area. Both datasets consist of aggregated and timely features related to eye fixations, saccades, blinks, and pupil diameters which are commonly used in VR/AR applications since they represent individual user visual behaviors. In addition, the state-of-the-art works are also evaluated using these datasets. That is why we evaluated our methods on
these datasets. We clarify this in the “Datasets” section of our paper. Furthermore, as we keep the signal sizes small by partitioning the data into chunks, function sensitivities decrease as well as the amount of required noise to obtain differential privacy. This operation can be applied independent of the dataset as long as temporal signals are used. Therefore, as we also mention in the “Conclusion” section of our paper, our methods could be applied to
other biometric related data as well.

Thank you very much for your comments!

Response to the Reviewer #2:

Reviewer #2: The authors are studying an interesting topic of privacy-preserving eye tracking data release, which is important given the rapid advancement of VR and AR technologies. However, this paper has a lot room to improve. Here I list several essential weak points in the hope that the authors could provide a better version of this work.

9) The motivation of this work needs to be clarified. For example, why eye movement data are sensitive?

Thank you very much for your request to clarify our motivations. Since it is possible to authenticate and identify individuals by using their eye movements data (because they have unique characteristics and they can leak private information such as whether an individual is going through depression, which can be detected by checking the eye-movement reaction time that is on average longer for a depressed person), they are considered as sensitive information. We motivate this in the
“Introduction” section of our paper (Please refer to the first paragraph of the “Introduction”.). 

10) Why DP is vulnerable under temporal correlation? (also see W3)

Thank you for your request to explain the vulnerability of DP mechanisms to correlations. As an example, we can consider that Alice has provided her eye movement feature signals. As illustrated for the publicly available datasets in our paper, such feature signals are highly correlated over time due to the nature of feature generation pipeline of eye movements (e.g., using windows sizes of 30 seconds with the step size of 0.5 and 1 seconds for MPIIDPEye and MPIIPrivacEye,
respectively.). As Alice’s eye movement data at time t=1 and t=2 are almost same [due to high correlation], adding Laplacian noise component (as an example DP mechanism) to each raw data point independently will not hide the sensitive information because the average of two noisy data points will be very close to the original raw values due to the symmetry in the Laplacian noise (Please refer especially to [25, 42, 43] in the paper. Refer to “Steil et al. 2019 [20]”
in the paper for the shortcomings of the standard differential privacy mechanisms on eye movements.). Taking these into consideration, in order to hide the sensitive information and provide differential privacy for eye movements, methods that take data correlation into consideration should be applied, such as our methods that aim at eliminating the correlations before applying any DP mechanism. We also clarify the motivation in the “Previous research” section.

11) Why it is reasonable to assume a trusted central server in this work? (basically, LDP is perforable than central DP since LDP has less trust assumption)

In most of the current VR/AR application use-cases, the user data, particularly eye tracking data, are collected by trusted entities and stored in a centralized fashion to model users’ behaviors to improve their product, as well as possibly due to the fact that VR/AR devices are not commonly available in the households. After having the complete data, various analyses are done such as user visual behaviors, gaze-guidance use-cases, foveated rendering, saliency estimation etc. by
these trusted entities. That is why we opted for a global differential privacy setting by assuming a trusted server. While being out of scope of this work, local differential privacy is surely a valid assumption with the scenario that each user sends their data to the server after applying local differential privacy mechanisms locally. However, for the eye tracking data collected from VR/AR setups, one should also guarantee that each user applies calibration procedures or similar steps
properly to obtain valid and high quality data, and there is already some research in that direction [1,2]. Overall, we foresee that both scenarios will go hand in hand in the near future regarding privacy-preservation since both setups are valid. For this work, we focus on the global differential privacy scenario that seems to fit well to the recent VR/AR applications. We clarify this at the end of the second paragraph of the “Introduction” section. Thank you!

12) The technical depth of the proposal method is limited. The proposed Chunk-based and difference-based methods seem trivial to me. The privacy guarantee is not clear; especially, how they could guarantee DP under temporal correlations. The authors may want to provide the formal proofs.

Thank you very much for your suggestion! We are very happy to provide further proofs, including a correction of the wrong result in the literature. In the revised manuscript, we provide the formal proof of FPA, which is our proof and which corrects the mistaken proof in the literature, and discuss the reasons why the chunk- and difference-based methods preserve the privacy via parallel and sequential compositions in CFPA and DCFPA sections, respectively. 

13) Important related works are missing. The following studies [a,b] demonstrate the vulnerability of DP under temporal correlations. The authors fail to acknowledge them and did not discuss whether or not the proposed methods in this work can address the vulnerability proven in [a,b].

[a] Quantifying Differential Privacy under Temporal Correlations, IEEE ICDE 2017.

[b] Quantifying Differential Privacy in Continuous Data Release Under Temporal Correlations. IEEE TKDE 2019.

Thank you very much for pointing out the missing related works. We have included them in our revised manuscript. These works mainly focus on the privacy leakage of standard differential privacy mechanisms and efficient ways to calculate the leakage in polynomial time. As we acknowledge such leakage of standard mechanisms, we have proposed low-complexity methods to reduce the data correlations to achieve differential privacy mechanisms that do not leak extra privacy due to the temporal
correlations. We clarify these in “Previous research”, “Chunk-based FPA”, and “Difference- and chunk-based FPA” sections. As illustrated in the “Correlation analysis” section, using our methods along with the FPA, temporal correlations reduce significantly thanks to the high decorrelation efficiency of the DFT and additional decorrelations that we provide with our methods. The privacy leakage due to the temporal correlations reduces
significantly by using the methods proposed in [a] and [b] as well. However, our approach is different from the one in these papers as in these papers the extra privacy loss for fixed differential privacy mechanisms is quantified when temporal correlations exist. Instead, we try to reduce the correlations in advance so that our proposed methods do not suffer from the privacy leakage due to the temporal correlations, which allows the use of standard differential privacy mechanisms for any
VR/AR dataset.

Thank you very much for your comments!

Response to the Reviewer #3:

Reviewer #3: Bozkir et al proposed a differential privacy method based on temporal signal processing methods to overcome possibly biomimetic information eavesdropping or information stealing. Their results focus on the correlation analysis of public available data in order to show different possible signal states. Then they apply a utility function in order to analyse how different techniques extended from the temporal signal processing methods will lead to less error in classifying the
signal states when noise is added to the raw signal, thus implementing the differential privacy scheme. The work is interesting and certainly has a timely contribution to this area. However, from the paper's current state, it is not clear whether there are significant benefits to privacy schemes of eye movement data.

14) The authors do not provide a direct comparison with existing methods in the literature, and most of the evaluated techniques are based on simple extension of temporal signal processing methods.

Thank you very much for your request! As reported in our paper, we have compared our methods/results with the state-of-the-art works in the eye tracking domain (Steil et al. 2019a [20], Steil et al. 2019b [28] in the paper) in terms of document-type, gender, and scene privacy sensitivity classifications from a practical point of view. In addition, we also compare our results with the standard Laplacian mechanism used to provide DP, and the corrected version of the FPA in terms of the
NMSE. The results show that we outperform both standard Laplacian mechanism and the FPA, the latter of which being suitable for time-series with temporal correlations. In addition, while hiding sensitive information such as personal identifiers and gender information with CFPA and DCFPA, document-type prediction and scene privacy sensitivity detection accuracies based solely on eye movements are close to the accuracies obtained from non-private mechanisms. Applying the FPA does not hide
personal identifiers as we are able to show that the person identification task works quite accurately after the application of the FPA. The strengths of CFPA and especially DCFPA stem from the significant decorrelation of the eye movement signals. We have strengthened our manuscript discussing these in a clearer way. In addition, in the literature, there are some works which do not formally provide privacy, which we have discussed in the “Previous research” section. Such
works rather apply simple mechanisms such as adding a pre-defined noise based on Gaussian distributions or spatial/temporal downsampling. Since the scope of such works are different compared to DP mechanisms that require mathematical privacy proofs, we have not directly compared their results with ours.

15) The metrics used for evaluating the privacy technique are focused on reconstruction of noisy data and its classification, but not entirely on how efficient in terms of privacy the proposed techniques really are.

In this work, we have focused on the epsilon-DP that is applied to eye movement feature signals. The reconstruction of eye movement signals is only one step in the proposed methods, which provide differential privacy. Therefore, the metric that we use to measure the privacy level is epsilon, which is the same for other differential privacy studies as well. In our study, the difference from the other traditional approaches is that observations of the eye movement signals are highly
correlated. Therefore, we aim to decrease the correlation before applying differential privacy mechanisms to the data. As also stated in various places in our paper, if the data are correlated, epsilon-prime is the actual privacy metric that should be used in practice since epsilon-prime is the differential privacy level that can be achieved against an attacker that is informed about the data correlations. In order to keep the epsilon-prime close to the epsilon value itself, we apply
various methods to decrease the data correlations. Thus, applying our methods the epsilon metric could be used along with the decorrelated data. We empirically show that data become decorrelated with the different sizes of chunks and difference (or subtraction) signals that are calculated. Overall, we discuss that there is a trade-off between the chunk size and remaining correlation, so the chunk size should be empirically decided for each dataset since the feature generation pipelines
could affect the temporal correlations positively or negatively in terms of privacy. We strengthen our arguments especially in the “Discussion” section based on your suggestions.

16) Moreover, in the last part of the paper, the authors spend a considerable amount of space on a totally different focus, which is classifying the different cohort features in the obtained datasets. This part has not a clear link to the differential privacy techniques and thus takes away a lot from the main message of the paper. I recommend the authors would make more space for evaluating further the proposed privacy methods directly.

We clarify why we additionally used classification accuracies for evaluation in the beginning of the “Results” section. In general, low normalized mean square error (NMSE) between original and private signals is favorable as this implies less variation from the original signals and this metric is analytically trackable. However, in practice, the private signals should be usable, i.e., different tasks should be applicable accurately. For this purpose, aligning with the
previous research on the privacy-preserving eye tracking, we have applied document-type prediction and scene privacy sensitivity detection tasks. In addition, again aligning with the previous work, we have analyzed whether gender prediction and person identification tasks could be applied accurately on the private signals, which is undesirable (David-John et al. 2021 [34], Steil et al. 2019 [20] in the paper). In general, while providing the differential privacy, it is important to hide
the personal information in the data and keep the data usable in practice. That is why we have used two utility metrics for evaluation and comparison, namely the NMSE metric and the classification accuracies. We show that with CFPA and DCFPA, while we keep the NMSE and person identification accuracies low, classification accuracies of document-type and scene privacy sensitivities high. On the contrary, the FPA method has both higher NMSE and higher person identification accuracies, which
implies it is still possible to infer significant amounts of personal information even though a high amount of noise is added. Since the decorrelation efficiency of the DCFPA is the highest, person identification accuracies are around the guessing probabilities for this method. Overall, we think that providing both metrics for evaluation and comparison gives a lot of insights about the privacy mechanisms and in line with the current literature. Therefore, we prefer keeping them in our
paper with an improved discussion, especially in the “Results” and “Discussion” sections.

17) Minor comments:

- We have updated the statement on HMDs and removed the argumentation on the cost of the devices as the cost itself is not necessarily relevant to our work due to the central nature of the global DP model we used (See the first paragraph of the “Introduction” section.).

- We have clarified the statement on adversarial behaviors (See the end of first paragraph and the beginning of second paragraph in the “Introduction” section.).

- We have already explained why we have chosen the LPA and the FPA in the third paragraph of the “Introduction” section and also in the “Fourier Perturbation Algorithm (FPA)” section. The LPA is the standard Laplace mechanism of differential privacy and FPA is one suitable mechanism for correlated data due to high decorrelation efficiency of the Discrete Fourier Transform (DFT) that is used in that algorithm.

- The confusing sentence is updated.

- When the DCFPA is used, the classification accuracies of document-type or privacy sensitivity tasks are kept in acceptable and usable ranges. At the same time, it is not possible to classify the participant information using the eye movement data when this method is applied in both datasets. Not being able to classify the participants is indeed a good thing which shows that also from a classification perspective, the personal information that eye movement features are hidden. Previous
work (Steil et al. 2019 [20] in the paper) also reports similar metrics for person identification. That is why, we think that it is quite relevant to our work and prefer to keep it in the paper. 

- We have updated the incomplete sentence.

Thank you very much for your comments!

References:

[1] E. Bozkir, S. Eivazi, M. Akgün and E. Kasneci, “Eye Tracking Data Collection Protocol for VR for Remotely Located Subjects using Blockchain and Smart Contracts,” 2020 IEEE International Conference on Artificial Intelligence and Virtual Reality (AIVR), 2020, pp. 397-401, doi: 10.1109/AIVR50618.2020.00083.

[2] X. Ma, M. Cackett, L. Park, E. Chien and M. Naaman, “Web-Based VR Experiments Powered by the Crowd,” 2018 World Wide Web Conference (WWW), 2018, pp. 33-43, doi:10.1145/3178876.3186034.

---

## [Decision Letter · Decision Letter 1]

7 Jul 2021

PONE-D-21-07774R1

Differential privacy for eye tracking with temporal correlations

PLOS ONE

Dear Dr. Bozkir,

Thank you for submitting your manuscript to PLOS ONE. After careful consideration, we feel that the paper has improved significantly and is quite close to meeting PLOS ONE’s publication criteria. We have a small number of outstanding points that we would like the authors to consider before publication. To speed up the process and in consideration of the reviewers' time, the paper won't be sent out to the reviewers again and instead a decision will be made by the editor
only.

A brief rebuttal letter that responds to each point raised by the academic editor and reviewer(s). You should upload this letter as a separate file labeled 'Response to Reviewers'.A marked-up copy of your manuscript that highlights changes made to the original version. You should upload this as a separate file labeled 'Revised Manuscript with Track Changes'.An unmarked version of your revised paper without tracked changes. You should upload this as a separate file labeled 'Manuscript'.

We look forward to receiving your revised manuscript.

Kind regards,

Luca Citi, PhD

Academic Editor

PLOS ONE

Journal Requirements:

Reviewers' comments:

Reviewer's Responses to Questions

**Comments to the Author**

1. If the authors have adequately addressed your comments raised in a previous round of review and you feel that this manuscript is now acceptable for publication, you may indicate that here to bypass the “Comments to the Author” section, enter your conflict of interest statement in the “Confidential to Editor” section, and submit your "Accept" recommendation.

Reviewer #1: All comments have been addressed

Reviewer #2: All comments have been addressed

Reviewer #3: All comments have been addressed

2. Is the manuscript technically sound, and do the data support the conclusions?

Reviewer #1: (No Response)

Reviewer #2: Yes

Reviewer #3: Yes

3. Has the statistical analysis been performed appropriately and rigorously? 

Reviewer #1: (No Response)

Reviewer #2: Yes

Reviewer #3: Yes

4. Have the authors made all data underlying the findings in their manuscript fully available?

Reviewer #1: (No Response)

Reviewer #2: Yes

Reviewer #3: Yes

5. Is the manuscript presented in an intelligible fashion and written in standard English?

Reviewer #1: (No Response)

Reviewer #2: Yes

Reviewer #3: Yes

6. Review Comments to the Author

Reviewer #1: (No Response)

Reviewer #2: I would agree with the acceptance of this paper since the authors nicely address my concerns with the previous version.

Reviewer #3: The authors have successfully addressed my comments. However, relevant information was found in the responses but not adequately included in the text. I recommend the authors to address the following points

- The introduction does not fully explain the work in a high level manner adequately. Some important motivation for parts of the work are still missing, namely a) the explanation about the motivation of the utility analysis is missing b) the lack of motivation for using the epsilon-DP as a privacy metric, even though used in other works, the reader needs to know why such metric.

- I recommend the authors to also think of a visual way to present their work, as it is not straightforward to understand how all the different privacy functions are used. I would also think of another way to present Algorithm 1, at the moment that algorithm seems to be not needed.

- In many answers in the response letter, the authors claim to have added more information in the discussion section. However, the additions made seem quite limited in terms of new information.

7. PLOS authors have the option to publish the peer review history of their article (what does this mean?). If published, this will include your full peer review and any attached files.

Reviewer #1: No

Reviewer #2: No

Reviewer #3: No

---

## [Author Response · Author response to Decision Letter 1]

23 Jul 2021

We thank the Academic Editor and the Reviewers for evaluating our revised manuscript along with our rebuttal and for their helpful suggestions that helped to improve the manuscript. As further points are requested only by Reviewer 3, in our rebuttal, we address the points raised by the Reviewer 3. We hope that all concerns have been addressed in a satisfying way.

Response to Reviewer 3:

1) The introduction does not fully explain the work in a high level manner adequately. Some important motivation for parts of the work are still missing, namely a) the explanation about the motivation of the utility analysis is missing b) the lack of motivation for using the epsilon-DP as a privacy metric, even though used in other works, the reader needs to know why such metric.

Thank you very much for the suggestions.

a) The utility analysis based on the metric normalized mean square error (NMSE) shows the trend of divergence of noisy aggregated signals (i.e., differentially private signals) from the original signals and this metric is analytically trackable. We have stated this in the first paragraphs of “Results” and “Classification accuracy results” sections. To make this point clearer upfront, we have added this motivation to the last paragraph of
“Introduction” section (Before “Previous research”) so that the reader can grasp the high level motivation before diving into technical details. 

b) As requested, we have further explained the differential privacy and motivated its usage for eye movements in the second paragraph of the “Introduction” section. More technical details are already available in “Materials and methods” section.

2) I recommend the authors to also think of a visual way to present their work, as it is not straightforward to understand how all the different privacy functions are used. I would also think of another way to present Algorithm 1, at the moment that algorithm seems to be not needed.

Thank you very much for the suggestions. We have added a figure to visually depict FPA (instead of the algorithmic representation of it), since it is a fundamental approach that we use throughout the manuscript. In addition, we have added a visual representation combining the CFPA and DCFPA as suggested. These figures could be seen as Figures 1 and 2, respectively.

3) In many answers in the response letter, the authors claim to have added more information in the discussion section. However, the additions made seem quite limited in terms of new information.

Thank you very much for your request. We discuss our findings in “Discussion” section as well as providing their implications. While our “Results” section focuses on experimental evaluations, their implications and comparisons with our initial expectations are discussed in “Discussion”. To deepen the understanding based on our findings and comparisons, we have added further points to our “Discussion” section, especially to the
first paragraph and strengthened the section by analyzing our findings in a more compact manner, especially by giving more details about the points mentioned in the previous revision and our previous rebuttal.

Thank you very much for your suggestions. We hope that all concerns have been addressed in a satisfying manner.

---

## [Editor Report · Decision Letter 2]

28 Jul 2021

Differential privacy for eye tracking with temporal correlations

PONE-D-21-07774R2

Dear Dr. Bozkir,

We’re pleased to inform you that your manuscript has been judged scientifically suitable for publication and will be formally accepted for publication once it meets all outstanding technical requirements.

Kind regards,

Luca Citi, PhD

Academic Editor

PLOS ONE

---

## [Editor Report · Acceptance letter]

5 Aug 2021

PONE-D-21-07774R2 

Differential privacy for eye tracking with temporal correlations 

Dear Dr. Bozkir:

I'm pleased to inform you that your manuscript has been deemed suitable for publication in PLOS ONE. Congratulations! Your manuscript is now with our production department. 

Kind regards, 

on behalf of

Dr. Luca Citi 

Academic Editor

PLOS ONE